# Noise-Aware System Identification for High-Dimensional Stochastic Dynamics

## Abstract

Stochastic dynamical systems are ubiquitous in physics, biology, and engineering, where both deterministic drifts and random fluctuations govern system behavior. Learning these dynamics from data is particularly challenging in high-dimensional settings with complex, correlated, or state-dependent noise. We introduce a noise-aware system identification framework that jointly recovers the deterministic drift and full noise structure directly from the trajectory data, without requiring prior assumptions on the noise model. Our method accommodates a broad class of stochastic dynamics, including colored and multiplicative noise, that scales efficiently to high-dimensional systems, and accurately reconstructs the underlying dynamics. Numerical experiments on diverse systems validate the approach and highlight its potential for data-driven modeling in complex stochastic environments.

## 1 Introduction

Stochastic differential equations (SDEs) provide a fundamental and versatile framework for modeling systems in which random fluctuations are intrinsic to the dynamics (Evans, 2013; Särkkä & Solin, 2019). Compared to deterministic ordinary differential equations (ODEs), SDEs incorporate noise explicitly–often through a Brownian motion term–allowing them to capture variability and uncertainty that strongly influence system behavior. This capability is essential for representing complex phenomena in physics, biology, chemistry, and finance, where stochasticity can be a dominant factor. By incorporating deterministic forces and random fluctuations in a unified mathematical description, SDEs offer a flexible modeling approach that is both theoretically rigorous and practically relevant.

We consider SDEs of the form

$$d\mathbf{x}_t = \boldsymbol{f}(\mathbf{x}_t)\,\mathrm{d}t + \sigma(\mathbf{x}_t)\,\mathrm{d}\mathbf{w}_t, \quad \mathbf{x}_t, \mathbf{w}_t \in \mathbb{R}^D,$$

where the drift $\boldsymbol{f} : \mathbb{R}^D \to \mathbb{R}^D$ and the diffusion coefficient $\sigma : \mathbb{R}^D \to \mathbb{R}^{D \times D}$ are potentially unknown. The driving noise $\mathbf{w}_t$ is a vector of independent standard Brownian motions. The noise structure of the SDE system is described by a state dependent covariance matrix $\Sigma : \mathbb{R}^D \to \mathbb{R}^{D \times D}$, where $\Sigma = \sigma\sigma^\intercal$. This general formulation encompasses many classical and modern models. In physics, the Langevin equation (Sachs et al., 2017; Coffey & Kalmykov, 2012; Ebeling et al., 2008; Talay, 2002) describes microscopic particle dynamics under both systematic forces and thermal fluctuations. In biology, stochastic Lotka–Volterra models (Takeuchi et al., 2006) capture population interactions in fluctuating environments, while other SDE-based models describe cellular processes and gene expression noise (Székely & Burrage, 2014; Dingli & Pacheco, 2011). In chemistry, the chemical Langevin equation (Wu et al., 2016) accounts for reaction kinetics in small-molecule regimes, where random molecular collisions cannot be neglected. In finance, SDEs form the basis of models such as Black–Scholes (Black & Scholes, 1973; Hull, 2017), Vasicek (Vasicek, 1977), and Heston (Heston, 1993), which incorporate uncertainty in asset prices, interest rates, and volatility. More recently, SDE formulations have emerged as the mathematical backbone of diffusion models in machine learning (Ho et al., 2020; Song et al., 2021), enabling state-of-the-art generative modeling methods.

Accurate application of SDEs requires careful calibration to empirical data so that both the deterministic drift and stochastic noise are faithfully represented. This is crucial for predictive power and for preserving physical interpretability. In many traditional settings, the functional forms of $\boldsymbol{f}$ and $\sigma$ are assumed known up to a small set of parameters, which can be estimated via least squares or

related regression techniques (Mrázek & Pospíšil, 2017; Abu-Mostafa, 2001). However, in modern applications—particularly those involving high-dimensional data where these functional forms are often unknown, and both the drift and the diffusion must be learned directly from the observed trajectories. Statistical inference for SDEs has a rich history (Kutoyants, 2004), with maximum-likelihood methods playing a central role when full trajectory data are available (Liptser & Shiryaev, 2001, Chapter 7). Recent advances have extended such methods to data-driven drift recovery Guo et al. (2024), but typically under restrictive noise assumptions, such as independence or constant variance.

In this work, we develop a noise-informed, trajectory-based learning framework for discovering the governing structures of SDEs directly from observational data. Unlike methods that estimate the drift alone or treat noise as a secondary effect, our approach embeds the noise process explicitly into the learning procedure and leverages information from the entire trajectory evolution, rather than focusing on isolated time points. This enables simultaneous recovery of both the drift $\boldsymbol{f}$ and the noise structure $\Sigma(\mathbf{x})$, including scalar or matrix-valued forms and fully state-dependent, correlated noise. We conduct a systematic investigation of the method's stability, accuracy, and computational efficiency across a variety of SDE models with different noise structures, demonstrating consistently superior performance in reconstructing complex stochastic dynamics.

The remainder of the paper is organized as follows. We discuss the general SDE model which our learning is based on in Section 2. Section 3 introduces the noise-informed likelihood formulation and the associated learning framework for recovering drift and noise. Section 4 presents numerical experiments on representative stochastic systems, highlighting accuracy and robustness across diverse noise settings. Section 5 concludes with a discussion of the implications, limitations, and potential extensions of our approach.

## 1.1 RELATED WORKS

System identification of the drift term from deterministic dynamics has been studied in many different scenarios, e.g. identification by enforcing sparsity such as SINDy (Brunton et al., 2016), neural network based methods such as NeuralODE (Chen et al., 2018), PINN (Raissi et al., 2019) and autoencoder (Xu et al., 2024), regression based Cucker & Smale (2002)p, and high-dimensional reduction variational framework (Lu et al., 2019). There are statistical methods which can be used to estimate the drift and noise terms using pointwise statistics. SINDy for SDEs was also developed in (Wanner & Mezić, 2024).

The observation data generated by SDEs can be treated as a time-series data with a mild assumption on the relationship between $\mathbf{x}_t$ and $\mathbf{x}_{t+\triangle t}$. Various deep neural network architectures can be used to learn the drift term as well as predicting the trajectory data, using RNN, LSTM, and Transformers, see (Liao et al., 2019; Yang et al., 2023; Wen et al., 2023) for detailed discussion.

However, most of these methods use a regression type of loss function defined as follows

$$\mathcal{E}_{\mathcal{H}}^{\text{Reg}}(\tilde{\boldsymbol{f}}) = \mathbb{E}\Big[\frac{1}{T}\int_{t=0}^{T} ||\tilde{\boldsymbol{f}}(\mathbf{x}_t) - \frac{\mathrm{d}\mathbf{x}_t}{\mathrm{d}t}||^2 \ \mathrm{d}t\Big].$$

Here the derivative $\frac{\mathrm{d}\mathbf{x}_t}{\mathrm{d}t}$ is loosely defined in the discrete sense (or weak sense). On the other hand, our likelihood induced loss of the form $\langle \tilde{\boldsymbol{f}}, \Sigma^{\dagger}\tilde{\boldsymbol{f}}\rangle \ \mathrm{d}t - 2\langle \tilde{\boldsymbol{f}}, \Sigma^{\dagger} \ \mathrm{d}\mathbf{x}_t\rangle$, is linked to the regression type loss through the expression

$$||\tilde{\boldsymbol{f}} - \frac{\mathrm{d}\mathbf{x}_t}{\mathrm{d}t}||^2 \ \mathrm{d}t = ||\tilde{\boldsymbol{f}}||^2 \ \mathrm{d}t - 2\langle \tilde{\boldsymbol{f}}, \mathrm{d}\mathbf{x}_t\rangle + ||\frac{\mathrm{d}\mathbf{x}_t}{\mathrm{d}t}||^2 \ \mathrm{d}t.$$

The major difference comes in the re-scaling by the noise and our loss is a derivation from a negative-log likelihood, which guarantees the existence and uniqueness of minimizers.

Furthermore, special high-dim drift terms living on low-dim manifolds with constant noise is investigated in (Lu et al., 2022); such loss is similar to ours when $\sigma(\mathbf{x}) = \sigma > 0$. In (Guo et al., 2024), a constant correlated noise matrix is studied.

## 2 MODEL EQUATION

Before introducing our learning framework for system identification from observed stochastic dynamics, we first establish the modeling setting and notation for the observational data. Let

$(\Omega, \mathcal{F}, (\mathbb{F}_t)_{0 \leq t \leq T}, \mathbb{P})$ be a filtered probability space, for a fixed and finite time horizon $T > 0$. As usual, the expectation operator with respect to $\mathbb{P}$ will be denoted by $\mathbb{E}_{\mathbb{P}}$ or simply $\mathbb{E}$. For random variables $X, Y$ we write $X \sim Y$, whenever $X, Y$ have the same distribution. We consider governing equations for stochastic dynamics of the following form

$$\mathrm{d}\mathbf{x}_t = \boldsymbol{f}(\mathbf{x}_t)\,\mathrm{d}t + \sigma(\mathbf{x}_t)\,\mathrm{d}\mathbf{w}_t, \quad \mathbf{x}_t, \mathbf{w}_t \in \mathbb{R}^D, \tag{1}$$

with some given initial condition $\mathbf{x}_0 \sim \mu_0$, here $\boldsymbol{f} : \mathbb{R}^D \to \mathbb{R}^D$ is the drift term, $\sigma : \mathbb{R}^D \to \mathbb{R}^{D \times D}$ is the diffusion coefficient. Without Loss of Generality, we assume that $\sigma$ is symmetric positive definite (SPD), i.e., $\sigma^{\mathsf{T}} = \sigma$, $\mathbf{x}^{\mathsf{T}}\sigma\mathbf{x} \geq 0$ with $\mathbf{x}^{\mathsf{T}}\sigma\mathbf{x} = 0$ iff $\mathbf{x} = \mathbf{0}$. Moreover, $\mathbf{w}$ represents a vector of independent standard Brownian motions. The covariance matrix of the SDE system is a symmetric positive definite matrix denoted by $\Sigma = \Sigma(\boldsymbol{x}) : \mathbb{R}^D \to \mathbb{R}^{D \times D}$ where $\Sigma = \sigma\sigma^{\mathsf{T}}$. We impose the following global regularity and growth conditions: there exist constants $C_1, C_2 > 0$ such that for all $\boldsymbol{x}, \boldsymbol{y} \in \mathbb{R}^D$

$$\begin{cases} \|\boldsymbol{f}(\boldsymbol{x}) - \boldsymbol{f}(\boldsymbol{y})\| + \|\sigma(\boldsymbol{x}) - \sigma(\boldsymbol{y})\|_{\mathrm{Fro}} \leq C_1 \|\boldsymbol{x} - \boldsymbol{y}\|, \\ \|\boldsymbol{f}(\boldsymbol{x})\|^2 + \|\sigma(\boldsymbol{x})\|_{\mathrm{Fro}}^2 \leq C_2\big(1 + \|\boldsymbol{x}\|^2\big). \end{cases}$$

Under these assumptions, equation 1 admits a unique strong solution $\{\mathbf{x}_t\}_{t \in [0,T]}$ adapted to the filtration $(\mathbb{F}_t)_{0 \leq t \leq T}$ for every square-integrable initial condition $\mathbf{x}_0 \sim \mu_0$.

## 3 LEARNING FRAMEWORK

We now introduce the methodology for learning the drift $\boldsymbol{f}$ and the diffusion $\sigma$ terms of stochastic differential equations from observed trajectory data. We assume continuous observation data $\{\mathbf{x}_t\}_{t \in [0,T]}$ for $\mathbf{x}_0 \sim \mu_0$, and that $\boldsymbol{f}$ and $\sigma$ are the only unknowns. We estimate these functions in two stages.

### 3.1 ESTIMATION OF THE DIFFUSION TERM

The diffusion coefficient $\sigma$ is first inferred using quadratic (co-)variation arguments. For two scalar stochastic processes $\mathrm{x}_t$ and $\mathrm{y}_t$, the quadratic variation over time interval $[0, T]$ is defined by

$$[\mathrm{x}_t, \mathrm{y}_t]_0^T = \lim_{|\Delta t_k| \to 0} \sum_{k=1}^{K} (\mathrm{x}(t_{k+1}) - \mathrm{x}(t_k))(\mathrm{y}(t_{k+1}) - \mathrm{y}(t_k)),$$

where $\Delta t_k = t_{k+1} - t_k$ and $\{0 = t_1 < t_2 < \cdots < t_K = T\}$ is a partition of the interval $[0, T]$. For a vector stochastic process $\mathbf{x}_t = [\mathrm{x}_1(t), \mathrm{x}_2(t), \ldots, \mathrm{x}_D(t)]^{\mathsf{T}}$, the quadratic variation matrix $[\mathbf{x}, \mathbf{x}]_0^T$ has entries $[\mathrm{x}_i(t), \mathrm{x}_j(t)]_0^T$ for $i, j = 1, \ldots, D$. Using such notation, the estimation of $\Sigma = \sigma\sigma^{\mathsf{T}}$ is the minimizer of the following loss function

$$\mathcal{E}_\sigma(\tilde{\Sigma}) = \mathbb{E}\Big[\big([\mathbf{x}_t, \mathbf{x}_t]_0^T - \int_{t=0}^{T} \tilde{\Sigma}(\mathbf{x}_t)\,\mathrm{d}t\big)^2\Big]. \tag{2}$$

Since $\sigma$ is SPD, $\sigma = \sqrt{\Sigma}$ is uniquely defined. If $\Sigma$ is constant, then the estimation can be simplified to $\tilde{\Sigma} = \frac{1}{T}\mathbb{E}\big[[\mathbf{x}_t, \mathbf{x}_t]_0^T\big]$. Note that estimation of $\Sigma$ does not dependent on the drift function $\boldsymbol{f}$.

### 3.2 ESTIMATION OF THE DRIFT TERM

Once $\Sigma$ is obtained, we estimate $\boldsymbol{f}$ by finding the minimizer to the following likelihood-based loss

$$\mathcal{E}_{\mathcal{H}}(\tilde{\boldsymbol{f}}) = \frac{1}{2}\mathbb{E}\Big[\int_{t=0}^{T} \langle \tilde{\boldsymbol{f}}(\mathbf{x}_t), \Sigma^{\dagger}(\mathbf{x}_t)\tilde{\boldsymbol{f}}(\mathbf{x}_t)\rangle\,\mathrm{d}t - 2\langle \tilde{\boldsymbol{f}}(\mathbf{x}_t), \Sigma^{\dagger}(\mathbf{x}_t)\,\mathrm{d}\mathbf{x}_t\rangle\Big], \tag{3}$$

where $\tilde{\boldsymbol{f}} \in \mathcal{H}$ with $\mathcal{H}$ being restricted to a convex and compact (w.r.t to $L_\infty$) function space determined by the observed data, $\langle \cdot, \cdot \rangle$ denotes the Euclidean inner product, and $\Sigma^{\dagger}$ is the pseudo-inverse of $\Sigma$, under our setting $\Sigma^{\dagger} = \Sigma^{-1}$. The differential $\mathrm{d}\mathbf{x}_t$ is approximated in practice by finite differences $\mathrm{d}\mathbf{x}_t \approx \mathbf{x}_{t+\Delta t} - \mathbf{x}_t$. This loss function arises from the Girsanov theorem and the Radon-Nikodym derivative for stochastic processes, see (Liptser & Shiryaev, 2001, Chpater 7) and Section 3.3 for details.

### 3.3 DERIVATION OF THE LOSS FOR THE DRIFT

We discuss the theoretical foundation of our methods in this section. Consider two Itô processes defined over measurable space $(\Omega, \mathcal{F})$ and let $\mathbb{P}_X$, $\mathbb{P}_Y$ be probability measures corresponding to processes $\mathbf{x}$ and $\mathbf{y}$, where

$$\mathrm{d}\mathbf{x}_t = \boldsymbol{f}(\mathbf{x}_t)\,\mathrm{d}t + \sigma(\mathbf{x}_t)\,\mathrm{d}\mathbf{w}_t,$$
$$\mathrm{d}\mathbf{y}_t = \boldsymbol{g}(\mathbf{y}_t)\,\mathrm{d}t + \sigma(\mathbf{y}_t)\,\mathrm{d}\mathbf{w}_t, \quad \mathbf{y}_0 = \mathbf{x}_0,$$

satisfying all assumptions in (Liptser & Shiryaev, 2001, Theorem 7.18) and its following corollary. Then, the Radon-Nikodym derivative, or the likelihood ratio, takes the form

$$\frac{\mathrm{d}\mathbb{P}_X}{\mathrm{d}\mathbb{P}_Y}(\mathbf{y}) = \exp\Big( \int_0^T \langle (\boldsymbol{f}_t - \boldsymbol{g}_t, \Sigma^\dagger\,\mathrm{d}\mathbf{y}_t \rangle - \frac{1}{2} \int_0^T \langle (\boldsymbol{f}_t - \boldsymbol{g}_t), \Sigma_t^\dagger(\boldsymbol{f}_t + \boldsymbol{g}_t \rangle\,\mathrm{d}t \Big), \tag{4}$$

where $\boldsymbol{f}_t = \boldsymbol{f}(\mathbf{y}_t)$, $\boldsymbol{g}_t = \boldsymbol{g}(\mathbf{y}_t)$, and $\Sigma_t = \Sigma(\mathbf{y}_t)$. Here let us assume that the observations are $\{\mathbf{x}_t\}_{t\in[0,T]}$. In view of the assumption of (Liptser & Shiryaev, 2001, Theorem 7.18), the $n$-dimensional adapted process $\Theta = \sigma^\dagger(\boldsymbol{f}(\mathbf{x}_t) - \boldsymbol{g}(\mathbf{x}_t))$ is such that $\int_0^T \|\Theta\|^2\,\mathrm{d}t < \infty$. By Girsanov theorem, $\widetilde{\mathbf{w}}_t = \mathbf{w}_t + \int_0^T \Theta_s\,\mathrm{d}s$ is an $n$-dimensional standard Brownian motion under probability measure $\mathbb{P}_Y$. Hence, $\mathrm{d}\mathbf{x}_t = \boldsymbol{f}(\mathbf{x}_t)\,\mathrm{d}t + \sigma(\mathbf{x}_t)(\mathrm{d}\tilde{\mathbf{w}}_t - \Theta_t\,\mathrm{d}t) = \boldsymbol{g}(\mathbf{x}_t)\,\mathrm{d}t + \sigma(\mathbf{x}_t)\,\mathrm{d}\tilde{\mathbf{w}}_t$. For convenience, we take $\boldsymbol{g} = 0$, in which case $\mathbf{x}_t$ becomes a Brownian process under $\mathbb{P}_Y$. Therefore $\mathbb{P}_Y(\{\mathbf{x}_t\}_{t\in[0,T]}|\boldsymbol{f})$ is now independent from $\boldsymbol{f}$ since $\mathbf{x}_t$ has no drift term under $\mathbb{P}_Y$. Putting such likelihood under the negative-log function, we arrive at our first loss as

$$\mathcal{E}_T(\tilde{\boldsymbol{f}}) = -\ln L(\boldsymbol{f}|\{\mathbf{x}_t\}_{t\in[0,T]}) = \int_0^T \big( \boldsymbol{f}(\mathbf{x}_t)^\intercal \Sigma^\dagger f(\mathbf{x}_t)\,\mathrm{d}t - 2\boldsymbol{f}(\mathbf{x}_t)^\intercal \Sigma^\dagger\,\mathrm{d}\mathbf{x}_t \big).$$

Here such loss function is used to handle observation data from one long trajectory (i.e. observed over large time), and it will be effective especially for ergodic systems. Moreover, we also consider the situation where multiple medium (or short-burst) trajectories with different initial conditions are observed, then we derive our loss function as the expectation (over trajectories with different initial conditions) of the negative-log-likelihood function as

$$\mathcal{E}(\tilde{\boldsymbol{f}}) = \mathbb{E}\big[ -\ln L(\boldsymbol{f}|\{\mathbf{x}_t\}_{t\in[0,T]}) \big] = \frac{1}{2}\mathbb{E}\Big[ \int_0^T \big( \boldsymbol{f}(\mathbf{x}_t)^\intercal \Sigma^\dagger f(\mathbf{x}_t)\,\mathrm{d}t - 2\boldsymbol{f}(\mathbf{x}_t)^\intercal \Sigma^\dagger\,\mathrm{d}\mathbf{x}_t \big) \Big].$$

### 3.4 CONVERGENCE THEOREM

We present the following convergence results in a theorem.

**Theorem 1.** *Given the continuous-time i.i.d trajectory data $\{\mathbf{x}_t^m\}_{m=1}^M$ for $t \in [0, T]$ and each $\mathbf{x}_t^m$ generated by equation 1, we define an estimator to $\boldsymbol{f}$ through minimizing the following loss*

$$\mathcal{E}_M(\tilde{\boldsymbol{f}}) = \frac{1}{2M} \sum_{m=1}^M \Big( \int_0^T \langle \tilde{\boldsymbol{f}}_t^m, (\Sigma_t^m)^{-1} \tilde{\boldsymbol{f}}_t^m \rangle\,\mathrm{d}t - 2 \int_0^T \langle \tilde{\boldsymbol{f}}_t^m, (\Sigma_t^m)^{-1}\,\mathrm{d}\mathbf{x}_t^m \rangle \Big),$$

*where $\tilde{\boldsymbol{f}}_t^m = \tilde{\boldsymbol{f}}(\mathbf{x}_t^m)$, $\Sigma_t^m = \Sigma(\mathbf{x}_t^m)$, and $\tilde{\boldsymbol{f}} \in \mathbb{H}$ with $\mathbb{H}$ being convex and compact (w.r.t to $L^2$-norm). When $\mathbb{H}$ is finite dimensional, i.e., $n = \dim(\mathbb{H}) < \infty$, and $\boldsymbol{f} \in \mathbb{H}$, then the estimator, given as $\hat{\boldsymbol{f}}_M = \arg\min_{\tilde{\boldsymbol{f}} \in \mathbb{H}} \mathcal{E}_M(\tilde{\boldsymbol{f}})$, has the following properties: $\hat{\boldsymbol{f}}_M \xrightarrow{P} \boldsymbol{f}$ (consistency) and $\sqrt{M}(\hat{\boldsymbol{f}}_M - \boldsymbol{f}) \xrightarrow{D} \mathcal{N}(\boldsymbol{0}, \boldsymbol{B}^{-1})$ (Asymptotic normality). Here $\boldsymbol{B} = \mathbb{E}[\int_0^T \Psi_t^\intercal \Sigma_t^{-1} \Psi_t\,dt]$, where*

$$\Psi_t = \Psi(\mathbf{x}_t) = [\boldsymbol{\psi}_1(\mathbf{x}_t) \quad \cdots \quad \boldsymbol{\psi}_n(\mathbf{x}_t)] \in \mathbb{R}^{D \times n}.$$

*with $\{\boldsymbol{\psi}_1, \boldsymbol{\psi}_2, \cdots, \boldsymbol{\psi}_n\}$ being a basis of $\mathbb{H}$ where each $\boldsymbol{\psi}_\eta : \mathbb{R}^D \to \mathbb{R}^D$. Notice that $\boldsymbol{B}$ is SPD.*

### 3.5 DEEP LEARNING FOR HIGH-DIM FUNCTIONS

In learning high dimensional $\boldsymbol{f}$ and $\sigma$, we can employ the deep learning architecture, with one neural network for learning $\boldsymbol{f}$ and the other for $\sigma$. The learning of $\boldsymbol{f}$ is rather straightforward, since the loss is well-defined for deep learning and simply changing the functional space to be a space of

neural networks. We will discuss the learning of $\sigma$ in details. Let $G : \mathbb{R}^D \to \mathbb{R}^{D(D+1)/2}$ be a neural network with outputs arranged as $\{\boldsymbol{u}_{ij}(\boldsymbol{x})\}_{1 \leq j \leq i \leq D}$. Since $\Sigma$ is SPD, the Cholesky decomposition on $\Sigma$ gives $\Sigma(\boldsymbol{x}) := L(\boldsymbol{x}) L(\boldsymbol{x})^\intercal$ where $L$ is a lower-triangular matrix with positive diagonal entries. Therefore we can learn a lower–triangular mapping $\tilde{L} : \mathbb{R}^D \to \mathbb{R}^{D \times D}$ by

$$\big(\tilde{L}(\boldsymbol{x})\big)_{ij} = \begin{cases} h\big(\boldsymbol{u}_{ii}(\boldsymbol{x})\big) & \text{if } i = j \\ \boldsymbol{u}_{ij}(\boldsymbol{x}) & \text{if } i > j \;, \\ 0 & \text{if } i < j \end{cases}$$

where $h : \mathbb{R} \to (0, \infty)$ is some chosen function to enforce positivity on the main diagonal. Hence, we define the model $\tilde{\Sigma}(\boldsymbol{x}) := \tilde{L}(\boldsymbol{x}) \tilde{L}(\boldsymbol{x})^\intercal \approx \Sigma(\boldsymbol{x})$. Given $M$ trajectories, set $Y_l^m := \frac{\Delta \mathbf{x}_l^m \big(\Delta \mathbf{x}_l^m\big)^\intercal}{\Delta t}$. We learn the estimator by minimizing the Frobenius mean squared difference between $Y_l$ and $\tilde{\Sigma}(\mathbf{x}_l)$ over all observed trajectories:

$$\mathcal{E}(\tilde{\Sigma}) = \frac{1}{M} \sum_{m=1}^{M} \sum_{l=0}^{L-1} \left\| Y_l^m - \tilde{\Sigma}\big(\mathbf{x}_l^m\big) \right\|_{\mathrm{F}}^2. \tag{5}$$

If $\sigma$ is a full matrix, we use the matrix-square-root function to obtain $\sigma = \sqrt{\Sigma}$. If $\Sigma(\mathbf{x})$ is diagonal for all $\boldsymbol{x}$, i.e., $\Sigma(\boldsymbol{x}) = \mathrm{diag}\big(\Sigma_{11}(\boldsymbol{x}), \ldots, \Sigma_{DD}(\boldsymbol{x})\big)$, then we will learn each diagonal entry by a single-output positive network. Writing $Y_{l,ii}^m = \frac{\big(\Delta \mathbf{x}_l^{(m,i)}\big)^2}{\Delta t}$, where $\mathbf{x}_l^{(m,i)}$ represents the $i^{th}$ entry of $\mathbf{x}_l^m$ and the loss function can be decoupled and become $\mathcal{E}(\tilde{\Sigma}_{ii}) = \frac{1}{M} \sum_{m=1}^{M} \sum_{l=0}^{L-1} \Big( Y_{l,ii}^m - \tilde{\Sigma}_{ii}\big(\mathbf{x}_l^m\big) \Big)^2$. Hence $\hat{\sigma}_{ii}(\boldsymbol{x}) = \sqrt{\hat{\Sigma}_{ii}(\boldsymbol{x})}$.

### 3.6 Performance Measures

In order to properly gauge the accuracy of our learning estimators, we provide three different performance measures of our estimated drift. First, if we have access to original drift function $\boldsymbol{f}$, then we will use the following error to compute the difference between $\hat{\boldsymbol{f}}$ (our estimator) to $\boldsymbol{f}$ with the following norm

$$||\boldsymbol{f} - \hat{\boldsymbol{f}}||_{L^2(\rho)}^2 = \int_{\mathbb{R}^d} ||\boldsymbol{f}(\mathbf{x}) - \hat{\boldsymbol{f}}(\mathbf{x})||_{\ell^2(\mathbb{R}^D)}^2 \; \mathrm{d}\rho(\mathbf{x}), \tag{6}$$

where the weighted measure $\rho$, defined on $\mathbb{R}^D$, is $\rho(\mathbf{x}) = \mathbb{E}\Big[\frac{1}{T} \int_{t=0}^{T} \delta_{\mathbf{x}_t}(\mathbf{x})\Big]$. Here $\mathbf{x}_t$ evolves from $\mathbf{x}_0$ by equation 1. The norm given by 6 is useful only from the theoretical perspective, e.g. showing convergence. Under normal circumstances, $\boldsymbol{f}$ is most likely non-accessible. Thus we look at a performance measure that compares the difference between $\mathbf{X}(\boldsymbol{f}, \mathbf{x}_0, T) = \{\mathbf{x}_t\}_{t \in [0,T]}$ (the observed trajectory that evolves from $\mathbf{x}_0 \sim \mu_0$ with the unknown $\boldsymbol{f}$) and $\hat{\mathbf{X}}(\hat{\boldsymbol{f}}, \mathbf{x}_0, T) = \{\hat{\mathbf{x}}_t\}_{t \in [0,T]}$ (the estimated trajectory that evolves from the same $\mathbf{x}_0$ with the learned $\hat{\boldsymbol{f}}$ and driven by the same realized random noise as used by the original dynamics). Then, the difference between the two trajectories is measured as follows

$$||\mathbf{X} - \hat{\mathbf{X}}|| = \mathbb{E}\Big[\frac{1}{T} \int_{t=0}^{T} ||\mathbf{x}_t - \hat{\mathbf{x}}_t||_{\ell^2(\mathbb{R}^D)}^2 \; \mathrm{d}t\Big]. \tag{7}$$

However, comparing two sets of trajectories (even with the same initial condition) on the same random noise is not realistic. Therefore we compare the distribution of the trajectories over different initial conditions and different noise at the same time instances using the Wasserstein distance at any given time $t \in [0, T]$. Let $\mu_t^M$ be the empirical distribution at time $t$ for the simulation under $\boldsymbol{f}$ with $M$ trajectories, and $\hat{\mu}_t^M$ be the empirical distribution at time $t$ for the simulation with $M$ trajectories under $\hat{\boldsymbol{f}}$, where $\mu_t^M = \frac{1}{M} \sum_{i=1}^{M} \delta_{\mathbf{x}^{(i)}(t)}, \hat{\mu}_t^M = \frac{1}{M} \sum_{i=1}^{M} \delta_{\hat{\mathbf{x}}^{(i)}(t)}$. Then the Wasserstein distance of order two between $\mu_t^M$ and $\hat{\mu}_t^M$ is defined as

$$\mathcal{W}_2(\mu_t^M, \hat{\mu}_t^M \,|\, \mu_0) = \left( \inf_{\pi \in \Pi(\mu_t^M, \hat{\mu}_t^M \,|\, \mu_0)} \int_{\mathbb{R}^D \times \mathbb{R}^D} ||x - y||^2 \; \mathrm{d}\pi(x, y) \right)^{1/2}. \tag{8}$$

Here, $\Pi(\mu_t^M, \hat{\mu}_t^M \,|\, \mu_0)$ is the set of all joint distributions on $\mathbb{R}^D \times \mathbb{R}^D$ with marginals $\mu_t^M$ and $\hat{\mu}_t^M$, and with the additional constraint that the joint distribution must be consistent with the initial distribution of $\mathbf{x}_0$ following $\mu_0$.

## 4  EXAMPLES

We demonstrate the application of our trajectory-based method for estimating drift functions and noise structures, showcasing a variety of examples. We focus on two major types of normal SDEs, interacting partial systems, and Stochastic Partial Differential Equations (SPDEs), where the dimension of the systems can increase rapidly.

### 4.1  EXAMPLE: INTERACTING PARTICLE SYSTEMS (IPS)

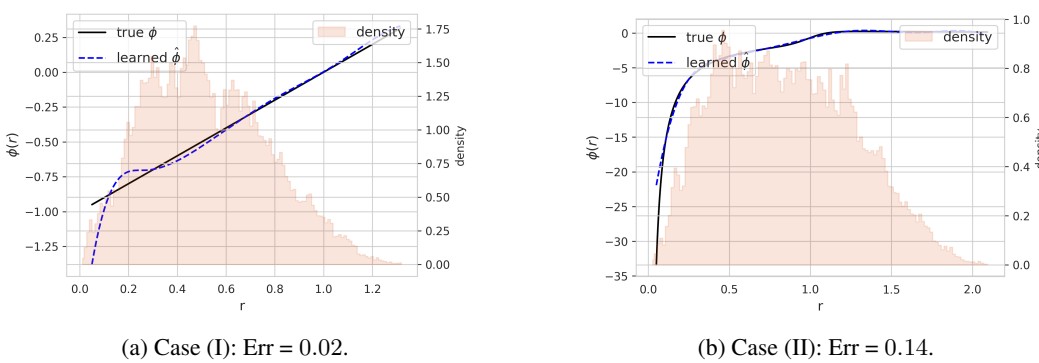

(a) Case (I): Err = 0.02.           (b) Case (II): Err = 0.14.

Figure 1: True $\phi$ vs learned $\hat{\phi}$; Empirical density of $r$ shown in the background.

We consider a high dimensional SDE case where the drift term has a special structure. Such special structure will allow us to learn the high-dimensional SDE more effectively through an innate dimension reduction approach. This high dimensional SDE case is a presentation of an interacting partial system. Learning of such systems without stochastic noise terms had been investigated in (Lu et al., 2019; Zhong et al., 2020; Maggioni et al., 2021; Feng et al., 2022; Feng & Zhong, 2024). We consider such system with correlated and state-dependent stochastic noise, i.e. for a system of $N$ particles, where each particle is associated with a state vector $\mathbf{x}_i \in \mathbb{R}^d$. The particles' states are governed by the following system of SDEs

$$\mathrm{d}\mathbf{x}_i(t) = \frac{1}{N} \sum_{j=1, j \neq i}^{N} \phi(\|\mathbf{x}_j(t) - \mathbf{x}_i(t)\|)(\mathbf{x}_j(t) - \mathbf{x}_i(t)) \, \mathrm{d}t + \sigma^{\mathrm{x}}(\mathbf{x}_i(t)) \, \mathrm{d}\mathbf{w}(t), \quad i = 1, \ldots, N.$$

Here $\phi : \mathbb{R}^+ \to \mathbb{R}$ is an interaction kernel that governs how partial $j$ influences the behavior of partial $i$, and $\sigma^{\mathrm{x}} : \mathbb{R}^d \to \mathbb{R}^{d \times d}$ is a symmetric positive definite matrix that represents the noise strength and correlation. We test two interaction kernels

$$\text{Case (I)}: \quad \phi(r) = r - 1,$$

$$\text{Case (II)}: \quad \phi(r) = -\frac{\tanh\big(8(1 - r)\big) + 0.67}{r}.$$

The diffusion is shared across particles, diagonal, and state–dependent, i.e., $\sigma^{\mathrm{x}}(\mathbf{x}_i(t)) = \mathrm{diag}\big(\sigma^{\mathrm{x}}_{11}(\mathbf{x}_i(t)), \sigma^{\mathrm{x}}_{22}(\mathbf{x}_i(t))\big)$ with

$$\begin{cases} \sigma^{\mathrm{x}}_{11}(\mathbf{x}_i(t)) &= 0.08 \sin^2\big(\|\mathbf{x}_i(t)\|\big) + \varepsilon, \\ \sigma^{\mathrm{x}}_{22}(\mathbf{x}_i(t)) &= 0.06 \cos^2\big(\|\mathbf{x}_i(t)\|\big) + \varepsilon, \end{cases} \quad \varepsilon = 0.01.$$

We run two experiments to justify our method. We take $N = 30$ particles in $\mathbb{R}^d$ with $d = 2$ (so $D = Nd = 60$), time horizon $T = 1$, step size $\Delta t = 0.001$, and $M = 100$ i.i.d. trajectories. The initial distributions are i.i.d. $\mathbf{x}_0 \sim \mathrm{Unif}([0,1]^d)$ for each particle. Simulation uses Euler–Maruyama method. In estimating $\sigma$, following the general implementation of $\sigma$ mentioned in  A.2, for the diagonal case we learn each diagonal entry independently. **Conclusion**: the comparison of the trajectories in Fig. 2a and 2b shows that the learned $\hat{\mathbf{x}}$ is close to the true $\mathbf{x}$ under the same noise. The comparison of $\phi$ vs $\hat{\phi}$ in Fig. 1a and 1b shows that when the data is abundant (the background shows the pairwise distance data used to obtain $\hat{\phi}$), the two are close to each other; for $r$ close to zero, due to

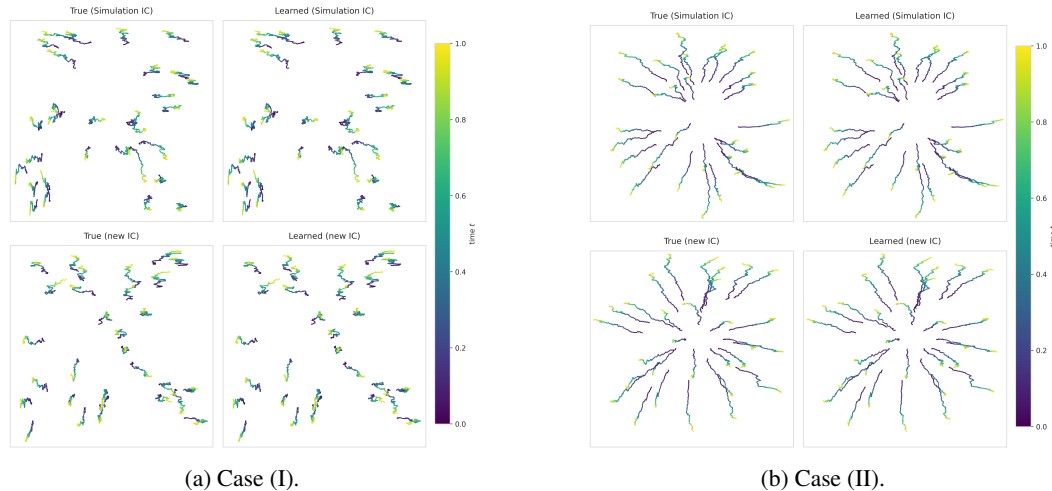

(a) Case (I).        (b) Case (II).

Figure 2: True $\mathbf{x}$ vs learned $\hat{\mathbf{x}}$ under the same noise. Top row: evolution from the same training IC. Bottom row: evolution from a new IC.

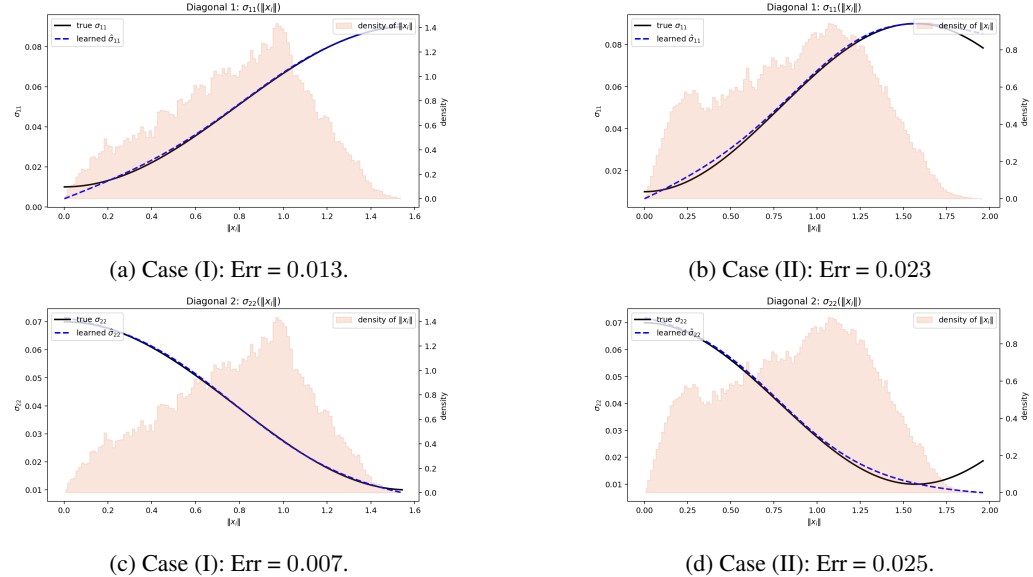

(a) Case (I): Err = 0.013.        (b) Case (II): Err = 0.023

(c) Case (I): Err = 0.007.        (d) Case (II): Err = 0.025.

Figure 3: True $\Sigma_{ii}$ vs learned $\hat{\Sigma}_{ii}$ for $i = 1, 2$.

the form of the system, i.e. $\phi(||\mathbf{x}_j - \mathbf{x}_i||)(\mathbf{x}_j - \mathbf{x}_i)$, the information is weighted by zero, our learning is not that promising. Figure 3 show our estimation result on state dependent $\sigma$ under two different kinds of dynamics. Each diagonal entry is modeled by a shallow two–hidden–layer Tanh network with width 32. The estimators tracks the true $\sigma$ closely even with such a lightweight network.

## 4.2 EXAMPLE: SPDE ESTIMATION

We extend our method of section 3 to the stochastic heat equation with additive noise

$$\mathrm{d}\mathbf{u}(t, \mathbf{x}) - \theta(\mathbf{x}) \, \Delta\mathbf{u}(t, \mathbf{x}) \, \mathrm{d}t = \sigma \, \mathrm{d}\mathbf{w}(t, \mathbf{x}), \tag{9}$$

on a smooth bounded domain $G \subset \mathbb{R}^d$, with initial condition $\mathbf{u}(0, \mathbf{x}) = 0$, zero boundary condition, and where $\Delta$ denotes the Laplace operator on $G$ with zero boundary conditions. The existence, uniqueness and other analytical properties of the solution $\mathbf{u}$ are well understood, and we refer to (Lototsky & Rozovsky, 2017). Throughout this section, we fix the Hilbert space $H = L^2(G)$

equipped with the usual inner product denoted by $(\cdot,\cdot)_H$. We note that in this case, the Laplace operator $\Delta$ has only point spectrum, and we denote by $\{h_k : k \in \mathbb{N}\} \subset H$ its eigenfunctions and $-\lambda_k$ the corresponding eigenvalues, i.e. $\Delta h_k = -\lambda_k h_k$. It is well known that $\{h_k : k \in \mathbb{N}\}$ is a complete system in $H$, and without loss of generality we assume it is also orthonormal. The space-time noise, is assumed to be a cylindrical Brownian motion in $H$, which informally can be written as $\mathbf{w}(t, \mathbf{x}) = \sum_{k \in \mathbb{N}} q_k h_k(\mathbf{x}) \mathbf{w}_k(t)$, where $\{q_k\}_{k \in \mathbb{N}} \subset (0, \infty)$ and $\{\mathbf{w}_k\}_{k \in \mathbb{N}}$ are independent one dimensional Brownian motions and $\sigma$ is a positive constant. Assume that $\theta$ is bounded, a.s. continuous on $G$, and $\theta(x) \geq c_0 > 0$, for some positive real $c_0$. This guarantees the existence of the solution to equation 9 in an appropriate triple of Hilbert spaces. We are interested in the estimation of $\theta(x)$. To verify our theoretical result, we present two numerical experiments for the stochastic

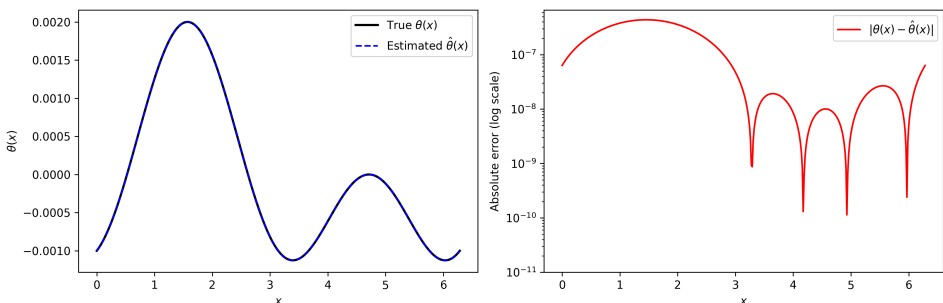

Figure 4: Left: Exact $\theta_1(x)$ (solid) vs $\hat{\theta}_1$ (dashed). Right: $|\theta_1 - \hat{\theta}_1|$ in log-scale.

heat equation equation 9 on the spatial interval $[0, 2\pi]$. Throughout we consider Fourier basis as our estimation function space, i.e., $\mathcal{H}_n = \mathrm{span}\{1, \sin(k\mathbf{x}), \cos(k\mathbf{x})\}_{k=1}^n$. In simulation of $\mathbf{u}(t, \mathbf{x})$, we apply a Galerkin projection of dimension $N_{\mathrm{full}}$ with time step $\Delta t = 10^{-3}$ up to horizon $T = 10$.

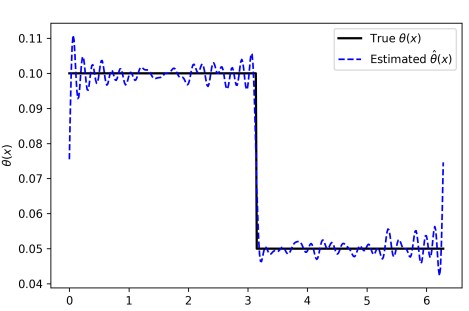

Figure 5: True $\theta_2$ (solid) vs $\hat{\theta}_2$ (dashed).

The drift matrix is calculated with true $\theta(x)$ where no projection error is introduced at the simulation stage. Only the first $N_{\mathrm{obs}}$ highest–frequency Fourier modes are marked as observable and the noise level is fixed at $\sigma = 0.2$. In the first experiment, we take $\theta_1(x) = 0.001 \left( \sin x - \cos 2x \right) \in \mathcal{H}_2$, set $N_{\mathrm{full}} = 100$ and $N_{\mathrm{obs}} = 40$. Next we consider a discontinuous coefficient outside the function space with

$$\theta_2(x) = \begin{cases} 0.10, & 0 \leq x < \pi \\ 0.05, & \pi \leq x \leq 2\pi \end{cases}$$

and set the estimation function subspace to $\mathcal{H}_{40}$. The simulation is carried out with $N_{\mathrm{full}} = 200$ and we observe $N_{\mathrm{obs}} = 100$ modes. Figure 4 and 5 show the effectiveness of our learning under two fundamentaly different scenarios, one with $\theta \in \mathcal{H}_n$ and the other with $\theta \notin \mathcal{H}_n$.

### 4.3 CONVERGENCE STUDY

To illustrate the statistical consistency of our estimator defined in 3, we consider an SDE in $d = 1$ case with $\boldsymbol{f}(\mathbf{x}) = -\mathbf{x}^3 + \mathbf{x}$ and $\sigma(\mathbf{x}) = 1 + 0.4 \sin \mathbf{x}$ simulated by the Euler–Maruyama scheme with step–size $\Delta t = 10^{-3}$. The initial states are drawn i.i.d. from the invariant density, so the process is strictly stationary. In $1D$, this density is

$$\pi(\mathbf{x}) = \frac{1}{G(\boldsymbol{f})} \, \sigma(\mathbf{x})^{-2} \exp\Big\{ 2 \int_0^{\mathbf{x}} \frac{\boldsymbol{f}(\mathbf{v})}{\sigma(\mathbf{v})^2} \, d\mathbf{v} \Big\}, \quad G(\boldsymbol{f}) = \int_{\mathbb{R}} \sigma(\mathbf{x})^{-2} \exp\Big\{ 2 \int_0^{\mathbf{x}} \frac{\boldsymbol{f}(\mathbf{v})}{\sigma(\mathbf{v})^2} \, d\mathbf{v} \Big\} \, d\mathbf{x}.$$

For a collection of $M$ paths observed over $[0, T]$ our drift estimator $\hat{\boldsymbol{f}}$ is searched in the space $\mathcal{H} = \mathrm{span}\{1, \mathbf{x}, \mathbf{x}^2, \mathbf{x}^3\}$ by minimizing the loss function 3. The estimation error is quantified in

the $\rho$–weighted norm introduced in equation 6. We test consistency in both time $T$ and number of observed trajectories $M$ with each replicated 20 times to obtain error bars. We first fix $M = 1$ and let $T \in \{4, 8, 16, 32, 64, 128\}$. Due to the ergodicity of the underlying SDE, we expect the following convergence rate $\|\hat{\boldsymbol{f}} - \boldsymbol{f}\|_{L^2(\rho)} = O(T^{-1/2})$, which is confirmed by Fig.6b. Next, we fix $T = 1$ and

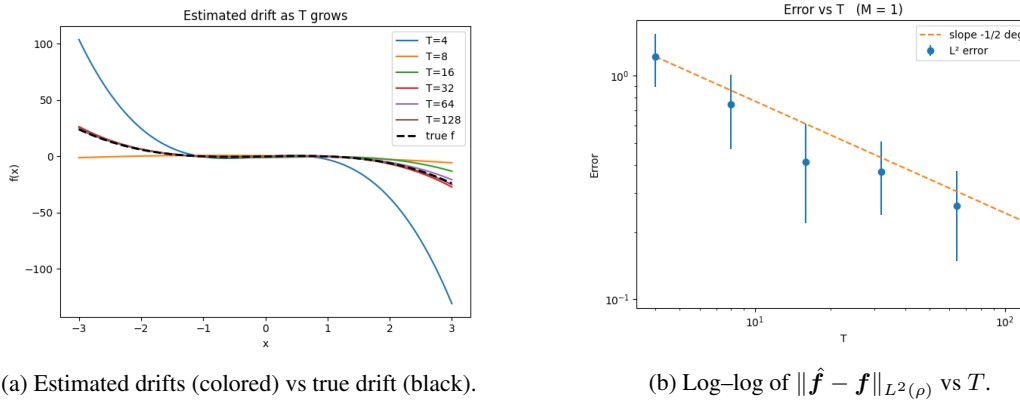

(a) Estimated drifts (colored) vs true drift (black).      (b) Log–log of $\|\hat{\boldsymbol{f}} - \boldsymbol{f}\|_{L^2(\rho)}$ vs $T$.

Figure 6: Convergence Test with $M = 1$.

let $M \in \{4, 8, 16, 32, 64, 128, 256\}$. The error decays at the rate $\|\hat{\boldsymbol{f}} - \boldsymbol{f}\|_{L^2(\rho)} = O(M^{-1/2})$, which is the rate confirmed by our theorem; see Fig.7b. In addition to Log–log plots 6b and 7b confirming

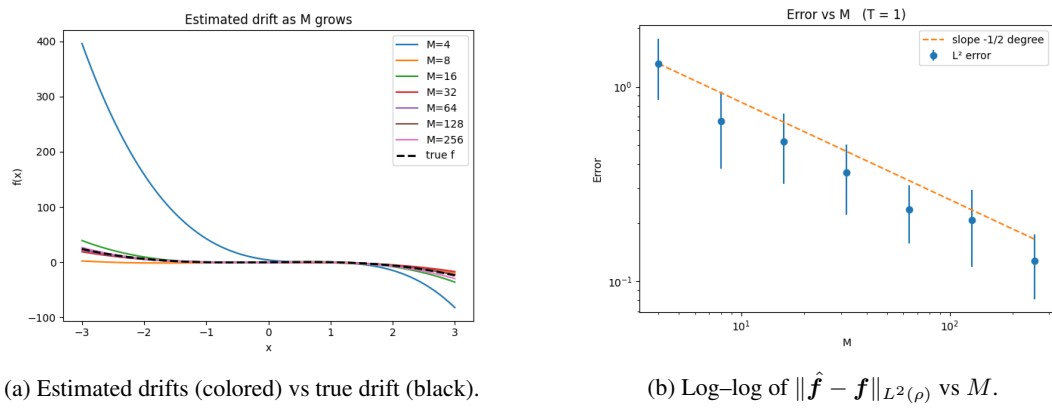

(a) Estimated drifts (colored) vs true drift (black).      (b) Log–log of $\|\hat{\boldsymbol{f}} - \boldsymbol{f}\|_{L^2(\rho)}$ vs $M$.

Figure 7: Convergence Test with $T = 1$.

the predicted slopes $-1/2$ in both regimes, we plot the corresponding drift functions 6a and 7a to illustrate the qualitative tightening of $\hat{\boldsymbol{f}}$ towards $\boldsymbol{f}$ as information increases. These numerical findings demonstrate that the estimator remains statistically consistent when the diffusion coefficient is state dependent.

## 5   CONCLUSION

We have demonstrated a novel learning methodology for inferring the drift and diffusion coefficient in general SDE systems driven by Brownian noise. Our estimation approach does not assume a specific functional structure for the drift or the diffusion, thereby enhancing its applicability across a diverse range of SDE models. This approach can handle high-dimensional SDE systems by leveraging deep learning architectures. The loss function for the drift is derived from the negative logarithm of the ratio of likelihood functions. For the diffusion coefficient, the loss function is based on the quadratic variation, which operates independently of the drift function. This independence makes our method particularly effective in scenarios where only trajectory observations are available. Additionally, our approach is adaptable to various noise structures.

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

## A  LEARNING FRAMEWORK

We discussion additional details related to the learning of drift and noise in this section.

### A.1  SIMPLIFICATION OF THE LOSS

When $D \gg 1$ and $\sigma = \sigma(\mathbf{x}) \in \mathbb{R}^{D \times D}$ is a full matrix, the learning of the drift term $\boldsymbol{f}$ can be computationally demanding, as all components of $\boldsymbol{f}$ are coupled and one has to solve the optimization problem in high-dimensional space all at once. Stochastic gradient descent coupled with neural network solutions is one of the desired approaches; however the solutions become less interpretable. In this section, we discuss several scenarios this loss for learning drift can be simplified. In this section, we discuss several scenarios in which the loss for learning drift can be simplified.

In the case of the noise being a constant full matrix, i.e. $\sigma(\mathbf{x}_t) = \sigma \in \mathbb{R}^{D \times D}$, the loss is equivalent (in the optimization sense) to the following

$$\mathcal{E}_{\mathcal{H}}^{\text{Sim}}(\tilde{\boldsymbol{f}}) = \mathbb{E}\Big[ \int_{t=0}^{T} ||\tilde{\boldsymbol{f}}(\mathbf{x}_t)||^2 \ \mathrm{d}t - 2\langle \tilde{\boldsymbol{f}}(\mathbf{x}_t), \mathrm{d}\mathbf{x}_t \rangle \Big]$$

In the case of state-dependent uncorrelated noise, i.e. $\Sigma(\boldsymbol{x}) = \sigma^2(\boldsymbol{x})\boldsymbol{I}$, where $\boldsymbol{I}$ is the $D \times D$ identity matrix and $\sigma : \mathbb{R}^D \to \mathbb{R}^+$ is a scalar function depending on the state and representing the noise level, the loss function equation 3 can be simplified to

$$\mathcal{E}_{\mathcal{H}}^{\text{Sim}}(\tilde{f}) = \mathbb{E}\Big[ \sum_{d=1}^{D} \int_{t=0}^{T} \frac{|\tilde{f}_d(\mathbf{x}_t)|^2 \ \mathrm{d}t - 2\tilde{f}_d(\mathrm{d}\mathbf{x})_d(t)}{2\sigma^2(\mathbf{x}_t)} \Big], \tag{10}$$

where $\tilde{\boldsymbol{f}}(\mathbf{x}_t) = (\tilde{f}_1(\mathbf{x}_t), \cdots, \tilde{f}_D(\mathbf{x}_t))$. Hence the learning of each component of $\boldsymbol{f}$ can be decoupled. When $\Sigma$ is a state-dependent full matrix, we consider the eigen-decomposition of $\Sigma$, i.e.

$\Sigma(\boldsymbol{x}) = \boldsymbol{Q}\boldsymbol{\Lambda}(\boldsymbol{x})\boldsymbol{Q}^{\intercal}$, then we rotate the system by $\boldsymbol{Q}^{\intercal}$, i.e., $\mathbf{x}'_t = \boldsymbol{Q}^{\intercal}\mathbf{x}_t$, $\boldsymbol{f}'(\boldsymbol{x}') = \boldsymbol{Q}^{\intercal}\boldsymbol{f}(\boldsymbol{x})\boldsymbol{Q}$, $\mathbf{w}'_t = \boldsymbol{Q}^{\intercal}\mathbf{w}_t$, then we obtain the case when $\Sigma$ is a diagonal matrix. Once we learn $\hat{\boldsymbol{\Lambda}}$ and $\boldsymbol{f}'$, we will use the following to obtain the original functions, i.e., $\boldsymbol{f}(\boldsymbol{x}) = \boldsymbol{Q}\boldsymbol{f}'(\boldsymbol{Q}\boldsymbol{x})\boldsymbol{Q}^{\intercal}$, and $\hat{\Sigma} = \boldsymbol{Q}\hat{\boldsymbol{\Lambda}}\boldsymbol{Q}^{\intercal}$.

## A.2 IMPLEMENTATION

We discuss in details how the algorithm is implemented for our learning framework. Practically speaking, data are rarely sampled continuously in time. Instead, observers typically have access to fragmented data sets, gathered from multiple independently sampled trajectories at specific, discrete time points $\{\mathbf{x}^m_l\}^{L,M}_{l,m=1}$, where $\mathbf{x}^m_l = \mathbf{x}^{(m)}(t_l)$ with $0 = t_1 < \cdots < t_L = T$ and $\mathbf{x}^m_0$ is an i.i.d sample from $\mu_0$. We use a discretized version of 3,

$$\mathcal{E}_{L,M,\mathcal{H}}(\tilde{\boldsymbol{f}}) = \frac{1}{2M}\sum_{l,m=1}^{L-1,M}\Big(\langle\tilde{\boldsymbol{f}}(\mathbf{x}^m_l), \Sigma^{-1}(\mathbf{x}^m_l)\tilde{\boldsymbol{f}}(\mathbf{x}^m_l)\rangle\Delta t_l - 2\langle\tilde{\boldsymbol{f}}(\mathbf{x}^m_l), \Sigma^{-1}(\mathbf{x}^m_l)\Delta\mathbf{x}^m_l\rangle\Big), \quad (11)$$

for $\tilde{\boldsymbol{f}} \in \mathcal{H}$ and $\Delta\mathbf{x}^m_l = \mathbf{x}^m_{l+1} - \mathbf{x}^m_l$. Moreover, we also assume that $\mathcal{H}$ is a finite-dimensional function space, i.e. $\dim(\mathcal{H}) = n < \infty$. Then for any $\tilde{\boldsymbol{f}} \in \mathcal{H}$, $\tilde{\boldsymbol{f}}(\boldsymbol{x}) = \sum_{i=1}^n \boldsymbol{a}_i\psi_i(\boldsymbol{x})$, where $\boldsymbol{a}_i \in \mathbb{R}^D$ is a constant vector coefficient and $\psi_i : \boldsymbol{D} \subset \mathbb{R}^D \to \mathbb{R}$ is a basis of $\mathcal{H}$ and the domain $\boldsymbol{D}$ is constructed by finding out the $\min/\max$ of the components of $\mathbf{x}_t \in \mathbb{R}^D$ for $t \in [0, T]$. We consider two methods for constructing $\psi_i$: $a$) use pre-determined basis such as piecewise polynomials or Clamped B-spline, Fourier basis, or a mixture of all of the aforementioned ones; $b$) use neural networks, where the basis functions are also trained from data. Next, we can put the basis representation of $\tilde{\boldsymbol{f}}$ back to equation 11, we obtain the following loss based on the coefficients

$$\mathcal{E}_{L,M,\mathcal{H}}(\{\boldsymbol{a}_\eta\}_{i=1}^n) = \frac{1}{2M}\sum_{l,m=1}^{L-1,M}\Big(\sum_{i,j=1}^n\langle\boldsymbol{a}_i, \Sigma^{-1}_{l,m}\boldsymbol{a}_j\rangle\psi^m_{i,l}\psi^m_{j,l}\Delta t_l - 2\sum_{i=1}^n\langle\boldsymbol{a}_i, \Sigma^{-1}_{l,m}\Delta\mathbf{x}^m_l\rangle\psi^m_{i,l}\Big),$$

$$(12)$$

where $\psi^m_{i,l} = \psi_i(\mathbf{x}^m_l)$, $\Sigma^{-1}_{l,m} = \Sigma^{-1}(\mathbf{x}^m_l)$ and $\Delta t_l = t_{l+1} - t_l$. In the case of diagonal covariance matrix $\Sigma$, i.e., $\Sigma(\boldsymbol{x}) = \text{diag}(\sigma^2_1(\boldsymbol{x}), \cdots, \sigma^2_D(\boldsymbol{x})) \in \mathbb{R}^{D\times D}$, for $\sigma_i > 0$ and $i = 1, \cdots, D$; we can re-write equation 12 as

$$\mathcal{E}_{L,M,\mathcal{H}}(\{\boldsymbol{a}_\eta\}_{i=1}^n) = \frac{1}{2M}\sum_{l,m=1}^{L-1,M}\Big(\sum_{i,j}^n\frac{\langle\boldsymbol{a}_i, \boldsymbol{a}_j\rangle}{\sigma^2_k(\mathbf{x}^m_l)}\psi^m_{i,l}\psi^m_{j,l}\Delta t_l - 2\sum_{i=1}^n\frac{\langle\boldsymbol{a}_i, \Delta\mathbf{x}^m_l\rangle}{\sigma^2_k(\mathbf{x}^m_l)}\psi^m_{i,l}\Big).$$

Here $(\boldsymbol{x})_k$ is the $k^{th}$ component of any vector $\boldsymbol{x} \in \mathbb{R}^D$. We define $\boldsymbol{\alpha}_k = [(\boldsymbol{a}_1)_k \quad \cdots \quad (\boldsymbol{a}_n)_k]^{\intercal} \in \mathbb{R}^n$, with $A_k \in \mathbb{R}^{n\times n}$ and $\boldsymbol{b}_k \in \mathbb{R}^n$ given as

$$A_k(i,j) := \frac{1}{2M}\sum_{l,m=1}^{L-1,M}\Big(\frac{\psi^m_{i,l}\psi^m_{j,l}}{\sigma^2_k(\mathbf{x}^m_l)}\Delta t_l\Big), \quad \boldsymbol{b}_k(i) := \frac{1}{2M}\sum_{l,m=1}^{L-1,M}\frac{\psi^m_{i,l}(\Delta\mathbf{x}^m_l)_k}{\sigma^2_k(\mathbf{x}^m_l)}.$$

Then the definition in (12) can be rewritten as $\mathcal{E}_{L,M,\mathcal{H}}(\{\boldsymbol{a}_\eta\}_{i=1}^n) = \sum_{k=1}^D(\boldsymbol{\alpha}^{\intercal}_k A_k\boldsymbol{\alpha}_k - 2\boldsymbol{\alpha}^{\intercal}_k\boldsymbol{b}_k)$. Since each $\boldsymbol{\alpha}^{\intercal}_k A_k\boldsymbol{\alpha}_k - 2\boldsymbol{\alpha}^{\intercal}_k\boldsymbol{b}_k$ is decoupled from each other, we just need to solve simultaneously $A_k\hat{\boldsymbol{\alpha}}_k - \boldsymbol{b}_k = 0$, for $k = 1, \ldots, D$. Then we can obtain $\hat{\boldsymbol{f}}(\boldsymbol{x}) = \sum_{i=1}^n\hat{\boldsymbol{a}}_i\psi_k(\boldsymbol{x})$. However when $\Sigma$ does not have a diagonal structure, we will have to resolve to gradient descent methods to minimize equation 12 in order to find the coefficients $\{\boldsymbol{a}_i\}_{i=1}^n$ for a total number of $nd$ parameters.

If a data-driven basis is desired, we set $\mathcal{H}$ to be the space of neural networks with the same depth, number of neurons, and activation functions in the hidden layers. Furthermore, we find $\hat{\boldsymbol{f}}$ by minimizing the loss given by the definition in (11) using any deep learning optimizer, such as Stochastic Gradient Descent or Adam, from well-known deep learning packages.

## A.3 PROOF OF THE THEOREM

We present the following definition about two different convergences of random variables.

**Definition 1.** *A sequence $\{x_1, x_2, \cdots, x_n\}$ of scalar random variables, with cumulative distribution functions, $\{F_1, F_2, \cdots, F_n\}$, is said to converge in distribution to a random variable $x$ with cumulative distribution function $F$ if*

$$\lim_{n \to \infty} F_n(x) = F(x),$$

*for every number $x \in \mathbb{R}$ at which $F$ is continuous. We denote such convergence as*

$$x_n \xrightarrow{D} x.$$

*We say $x_n$ convergences to $x$ in probability if for any $\epsilon > 0$, we have*

$$\lim_{n \to \infty} \mathbb{P}(|x_n - x| > \epsilon) = 0.$$

*We denote such convergence as*

$$x_n \xrightarrow{P} x.$$

The following lemma is needed for the convergence theorem.

**Lemma 1.** *Consider the space $(\mathbb{S}_{++}^n, \|\cdot\|_F)$ with $\mathbb{S}_{++}^n$ being the set of all $n \times n$ SPD matrices and $\|\cdot\|_F$ denoting the Frobenius norm, then the inversion map $g : \mathbb{S}_{++}^n \to \mathbb{S}_{++}^n$ defined by $g(\boldsymbol{A}) = \boldsymbol{A}^{-1}$ for $\boldsymbol{A} \in \mathbb{S}_{++}^n$ is continuous.*

*Proof.* For any $\boldsymbol{A} \in \mathbb{S}_{++}^n$ with $\det(\boldsymbol{A}) > 0$, we have

$$\boldsymbol{A}^{-1} = \frac{\text{adj}(\boldsymbol{A})}{\det(\boldsymbol{A})},$$

where $\text{adj}(\boldsymbol{A})$ is the adjugate matrix of $A$. Each entry of $\text{adj}(\boldsymbol{A})$ is a polynomial in the entries of $\boldsymbol{A}$, and $\det(\boldsymbol{A})$ is also a polynomial in the entries of $\boldsymbol{A}$. Since polynomials are continuous, both maps $\boldsymbol{A} \mapsto \text{adj}(\boldsymbol{A})$ and $\boldsymbol{A} \mapsto \det(\boldsymbol{A})$ are continuous on $\mathbb{R}^{n \times n}$. For $\boldsymbol{A} \in \mathbb{S}_{++}^n$, we have $\det(\boldsymbol{A}) > 0$, so the map $\boldsymbol{A} \mapsto \frac{\text{adj}(\boldsymbol{A})}{\det(\boldsymbol{A})}$ is continuous at $\boldsymbol{A}$ as the composition of continuous functions. Therefore, $g$ is continuous on $\mathbb{S}_{++}^n$. $\qquad\square$

We present the following uniform law of large numbers theorem. For the proof, please see (Newey & McFadden, 1994).

**Theorem 2** (Uniform Law of Large Numbers (Newey & McFadden, 1994))**.** *Let $\{x_i\}_{i=1}^\infty$ be i.i.d. and let $f(x, \theta)$ be some function defined for $\theta \in \Theta$. Assume:*

1. *$\Theta$ is compact;*

2. *for almost every $x$, the map $\theta \mapsto f(x, \theta)$ is continuous on $\Theta$, and for each $\theta \in \Theta$ the map $x \mapsto f(x, \theta)$ is measurable;*

3. *there exists a dominating function $h$ such that $\mathbb{E}[h(x)] < \infty$ such that $\|f(x, \theta)\| \leq h(x)$ for all $\theta \in \Theta$.*

*Then $\theta \mapsto \mathbb{E}[f(x, \theta)]$ is continuous in $\theta$ and*

$$\sup_{\theta \in \Theta} \|\frac{1}{n} \sum_{i=1}^n f(x_i, \theta) \; - \; \mathbb{E}[f(x, \theta)]\| \xrightarrow{P} 0.$$

The following theorem is needed to show convergence of vector-valued random variables. For the proof, please see (Vaart, 1998).

**Theorem 3** (Theorem 5.9 in (Vaart, 1998))**.** *Let $\Psi_n : \Theta \to \mathbb{R}^k$ be random vector–valued functions and $\Psi : \Theta \to \mathbb{R}^k$ a fixed vectored valued function of $\theta$. Suppose that for every $\varepsilon > 0$:*

$$\sup_{\theta \in \Theta} \|\Psi_n(\theta) - \Psi(\theta)\| \xrightarrow{P} 0, \qquad \inf_{\theta: \, \|\theta - \theta_0\| \geq \varepsilon} \|\Psi(\theta)\| \; > \; 0 \; = \; \|\Psi(\theta_0)\|.$$

*Then any sequence of estimator $\hat{\theta}_n$ such that $\Psi_n(\hat{\theta}_n) = o_p(1)$ converges in probability to $\theta_0$.*

We are now ready to show the proof of the convergence theorem.

*Proof.* We need to introduce a few quantities before we can establish the proof. First, we introduce the continuous form of $\mathcal{E}_M$. As $M \to \infty$, by law of large numbers, we have

$$\lim_{M \to \infty} \mathcal{E}_M(\tilde{\boldsymbol{f}}) = \mathcal{E}_\infty(\tilde{\boldsymbol{f}}) = \frac{1}{2}\mathbb{E}\Big[\int_0^T \langle \tilde{\boldsymbol{f}}_t, (\Sigma_t)^{-1}\tilde{\boldsymbol{f}}_t \rangle \, \mathrm{d}t - 2\int_0^T \langle \tilde{\boldsymbol{f}}_t, (\Sigma_t)^{-1} \, \mathrm{d}\mathbf{x}_t \rangle \Big],$$

where $\tilde{\boldsymbol{f}}_t = \tilde{\boldsymbol{f}}(\mathbf{x}_t)$, $\Sigma_t = \Sigma(\mathbf{x}_t)$. When $\mathbb{H}$ is finite dimensional, then for any $\tilde{\boldsymbol{f}} \in \mathbb{H}$, we have

$$\tilde{\boldsymbol{f}}(\boldsymbol{x}) = \sum_{\eta=1}^n \alpha_\eta \psi_\eta(\boldsymbol{x}) = \Psi(\boldsymbol{x})\boldsymbol{\alpha}, \quad \boldsymbol{\alpha} = \begin{bmatrix} \alpha_1 \\ \vdots \\ \alpha_n \end{bmatrix}.$$

Therefore, the two losses can be re-written as

$$\mathcal{E}_M(\tilde{\boldsymbol{f}}) = \frac{1}{2M}\sum_{m=1}^M \Big(\int_0^T (\Psi_t^m \boldsymbol{\alpha})^{\mathsf{T}}(\Sigma_t^m)^{-1}\Psi_t^m \boldsymbol{\alpha} \, dt - 2\int_0^T (\Psi_t^m \boldsymbol{\alpha})^{\mathsf{T}}(\Sigma_t^m)^{-1} \, \mathrm{d}\mathbf{x}_t^m\Big),$$

$$\mathcal{E}_\infty(\tilde{\boldsymbol{f}}) = \frac{1}{2}\mathbb{E}\Big[\int_0^T (\Psi_t \boldsymbol{\alpha})^{\mathsf{T}}(\Sigma_t)^{-1}\Psi_t \boldsymbol{\alpha} \, dt - 2\int_0^T (\Psi_t \boldsymbol{\alpha})^{\mathsf{T}}(\Sigma_t)^{-1} \, \mathrm{d}\mathbf{x}_t\Big),$$

Abusing the notation, we will use $\mathcal{E}_M(\tilde{\boldsymbol{f}})$ and $\mathcal{E}_M(\boldsymbol{\alpha})$ interchangeably; similarly for $\mathcal{E}_\infty(\tilde{\boldsymbol{f}})$ and $\mathcal{E}_\infty(\boldsymbol{\alpha})$, since $\boldsymbol{\alpha}$ and $\tilde{\boldsymbol{f}}$ have a one-on-one correspondence once a $\mathbb{H}$ is chosen.

Next, we will assume the following

$$\begin{cases} \mathbb{E}[\int_0^T \|\Psi_t^{\mathsf{T}}\Sigma_t^{-1}\Psi_t\|_2 \, dt] < \infty, \\ \mathbb{E}[\int_0^T \|\Psi_t^{\mathsf{T}}\Sigma_t^{-1}\boldsymbol{f}(\mathbf{x}_t)\|_2 \, dt] < \infty, \\ \mathbb{E}[\int_0^T \|\Psi_t^{\mathsf{T}}\sigma_t^{-1}\|_2 \, dt] < \infty, \end{cases}$$

Differentiating $\mathcal{E}_M$ w.r.t to $\boldsymbol{\alpha}$ gives

$$\nabla_{\boldsymbol{\alpha}}\mathcal{E}_M(\boldsymbol{\alpha}) = \frac{1}{M}\sum_{m=1}^M \Big(\int_0^T (\Psi_t^m)^{\mathsf{T}}(\Sigma_t^m)^{-1}(\Psi_t^m \boldsymbol{\alpha} \, \mathrm{d}t - \mathrm{d}\mathbf{x}_t^m)\Big).$$

Let

$$\phi_m(\boldsymbol{\alpha}) := \int_0^T (\Psi_t^m)^{\mathsf{T}}(\Sigma_t^m)^{-1}(\Psi_t^m \boldsymbol{\alpha} \, \mathrm{d}t - \mathrm{d}\mathbf{x}_t^m),$$

$$= \int_0^T (\Psi_t^m)^{\mathsf{T}}(\Sigma_t^m)^{-1}(\tilde{\boldsymbol{f}}_t^m \, \mathrm{d}t - \boldsymbol{f}_t^m \, \mathrm{d}t - \sigma_t^m \, \mathrm{d}\mathbf{w}_t^m),$$

$$= \int_0^T (\Psi_t^m)^{\mathsf{T}}(\Sigma_t^m)^{-1}(\tilde{\boldsymbol{f}}_t^m - \boldsymbol{f}_t^m) \, \mathrm{d}t - \int_0^T (\Psi_t^m)^{\mathsf{T}}(\sigma_t^m)^{-1} \, \mathrm{d}\mathbf{w}_t^m.$$

and define $\Phi_M(\boldsymbol{\alpha}) := \frac{1}{M}\sum_{m=1}^M \phi_m(\boldsymbol{\alpha})$. First, by Itô's formula

$$\mathbb{E}\Big[\int_0^T (\Psi_t^m)^{\mathsf{T}}(\sigma_t^m)^{-1} \, \mathrm{d}\mathbf{w}_t^m\Big] = \mathbf{0}.$$

Then

$$\mathbb{E}[\phi_m(\boldsymbol{\alpha})] = \mathbb{E}\Big[\int_0^T (\Psi_t^m)^{\mathsf{T}}(\Sigma_t^m)^{-1}(\tilde{\boldsymbol{f}}_t^m - \boldsymbol{f}_t^m) \, \mathrm{d}t\Big],$$

$$= \mathbb{E}\Big[\int_0^T \Psi_t^{\mathsf{T}}\Sigma_t^{-1}(\tilde{\boldsymbol{f}}_t - \boldsymbol{f}_t) \, \mathrm{d}t\Big]$$

Define

$$\Phi_\infty(\boldsymbol{\alpha}) = \lim_{m \to \infty} \Phi_M(\boldsymbol{\alpha}) = \mathbb{E}[\phi_m(\boldsymbol{\alpha})].$$

By theorem 2, since $\mathbb{H}$ is compact, $\phi_m$ is continuous at each $\boldsymbol{\alpha}$ and it is also bounded (by one of our assumptions). Moreover

$$\sup_{\tilde{\boldsymbol{f}} \in \mathbb{H}} ||\Phi_M(\boldsymbol{\alpha}) - \Phi_\infty(\boldsymbol{\alpha})|| = \sup_{\tilde{\boldsymbol{f}} \in \mathbb{H}} ||\frac{1}{M} \sum_{m=1}^M \phi_m(\boldsymbol{\alpha}) - \mathbb{E}[\Phi_m(\boldsymbol{\alpha})]|| \xrightarrow{P} 0.$$

Since $\boldsymbol{f} \in \mathbb{H}$, then $\boldsymbol{f}(\boldsymbol{x}) = \Psi(\boldsymbol{x})\boldsymbol{\alpha}_f$, then

$$\Phi_\infty(\boldsymbol{\alpha}) = \mathbb{E}[\int_0^T \Psi_t^\intercal \Sigma_t^{-1}(\tilde{\boldsymbol{f}}_t - \boldsymbol{f}_t) \, dt],$$

$$= \mathbb{E}[\int_0^T \Psi_t^\intercal \Sigma_t^{-1}(\Psi_t \boldsymbol{\alpha} - \Psi_t \boldsymbol{\alpha}_f) \, dt],$$

$$= \mathbb{E}[\int_0^T \Psi_t^\intercal \Sigma_t^{-1} \Psi_t \, dt](\boldsymbol{\alpha} - \boldsymbol{\alpha}_f)$$

$$= \boldsymbol{A}(\boldsymbol{\alpha} - \boldsymbol{\alpha}_f).$$

Since $\boldsymbol{A}$ is SPD, Let $\lambda_{\min}(\boldsymbol{A}) > 0$ be the minimal eignevalue of $\boldsymbol{A}$, then for all $\tilde{\boldsymbol{f}} \in \mathbb{H}$,

$$||\Phi_\infty(\boldsymbol{\alpha})|| = ||\boldsymbol{A}(\boldsymbol{\alpha} - \boldsymbol{\alpha}_f)|| \geq \lambda_{\min}(\boldsymbol{A})||\boldsymbol{\alpha} - \boldsymbol{\alpha}_f||.$$

Therefore, for any $\epsilon > 0$, we have

$$\inf_{||\boldsymbol{\alpha} - \boldsymbol{\alpha}_f|| \geq \epsilon} ||\Phi_\infty(\boldsymbol{\alpha})|| \geq \inf_{||\boldsymbol{\alpha} - \boldsymbol{\alpha}_f|| \geq \epsilon} \lambda_{\min}(\boldsymbol{A})||\boldsymbol{\alpha} - \boldsymbol{\alpha}_f|| \geq \lambda_{\min}(\boldsymbol{A})\epsilon > 0,$$

observe that $\Phi_\infty(\boldsymbol{\alpha}_f) = \boldsymbol{0}$. By theorme 3, we conclude that

$$\hat{\boldsymbol{f}}_M \xrightarrow{P} \boldsymbol{f}, \quad \text{convergence in probability.}$$

Next, recall

$$\Phi_M(\boldsymbol{\alpha}) = \frac{1}{M} \sum_{m=1}^M \int_0^T (\Psi_t^m)^\intercal (\Sigma_t^m)^{-1}(\Psi_t^m \boldsymbol{\alpha} \, dt - d\mathbf{x}_t^m),$$

define

$$\boldsymbol{A}_M = \frac{1}{M} \sum_{m=1}^M \int_0^T (\Psi_t^m)^\intercal (\Sigma_t^m)^{-1} \Psi_t^m \, dt.$$

Since $\boldsymbol{f}(\boldsymbol{x}) = \Psi(\boldsymbol{x})\boldsymbol{\alpha}_f$, hence

$$\phi_m(\boldsymbol{\alpha}_f) = \int_0^T (\Psi_t^m)^\intercal (\Sigma_t^m)^{-1}(\Psi_t^m \boldsymbol{\alpha}_f \, dt - d\mathbf{x}_t^m),$$

$$= \int_0^T (\Psi_t^m)^\intercal (\Sigma_t^m)^{-1}(\boldsymbol{f}_t^m \, dt - d\mathbf{x}_t^m),$$

$$= - \int_0^T (\Psi_t^m)^\intercal (\Sigma_t^m)^{-1} \sigma_t^m \, d\mathbf{w}_t^m,$$

$$= - \int_0^T (\Psi_t^m)^\intercal (\sigma_t^m)^{-1} \, d\mathbf{w}_t^m$$

This Itô integral is square-integrable, and $\mathbb{E}[\phi_m(\boldsymbol{\alpha}_f)] = \boldsymbol{0}$, and by Itô isometry

$$\text{Var}(\phi_m(\boldsymbol{\alpha}_f)) = \mathbb{E}[\int_0^T \Psi_t^\intercal \Sigma_t^{-1} \Psi_t \, dt] = \boldsymbol{A} < \infty.$$

Sincr $\mathbf{x}_t^m$ is i.i.d, $\phi_m(\boldsymbol{\alpha}_f)$ is also i.i.d. Therefore, by the multivariate Central Limit Theorem, we have

$$\sqrt{M}\Phi_M(\boldsymbol{\alpha}_f) = \frac{1}{\sqrt{M}} \sum_{m=1}^M \phi_m(\boldsymbol{\alpha}_f) \xrightarrow{D} \mathcal{N}(\boldsymbol{0}, \boldsymbol{A}).$$

Furthermore, we also have the following (recall $\hat{\boldsymbol{f}}(\boldsymbol{x}) = \Psi(\boldsymbol{x})\hat{\boldsymbol{\alpha}}$)

$$\Phi_M(\hat{\boldsymbol{\alpha}}) - \Phi_M(\boldsymbol{\alpha}_f) = \boldsymbol{A}_M(\hat{\boldsymbol{\alpha}} - \boldsymbol{\alpha}_f),$$

Since $\Phi_M(\hat{\boldsymbol{\alpha}}) = \boldsymbol{0}$, we obtain

$$\sqrt{M}(\hat{\boldsymbol{\alpha}} - \boldsymbol{\alpha}_f) = \sqrt{M}\boldsymbol{A}_M^{-1}\Phi_M(\boldsymbol{\alpha}_f).$$

Each entry of $\boldsymbol{A}_M$ is square-integrable and by law of large numbers $\boldsymbol{A}_M \to \boldsymbol{A}$ as $M \to \infty$ in probability entrywise, hence

$$||\boldsymbol{A}_m - \boldsymbol{A}||_F \xrightarrow{P} 0.$$

By lemma 1, the inversion mapping is continuous, hence

$$\boldsymbol{A}_M^{-1} \xrightarrow{P} \boldsymbol{A}^{-1}.$$

Putting them all together and by Slutsky's theorem, we end up with

$$\sqrt{M}(\hat{\boldsymbol{\alpha}} - \boldsymbol{\alpha}_f) \xrightarrow{D} \mathcal{N}(\boldsymbol{0}, \boldsymbol{A}^{-1}).$$

Furthermore, for a fixed $\boldsymbol{x}$, since $\hat{\boldsymbol{f}}(\boldsymbol{x}) = \Psi(\boldsymbol{x})\hat{\boldsymbol{\alpha}}$ and $\boldsymbol{f}(\boldsymbol{x}) = \Psi(\boldsymbol{x})\boldsymbol{\alpha}_f$, we finally have

$$\sqrt{M}(\hat{\boldsymbol{f}} - \boldsymbol{f}) \xrightarrow{D} \mathcal{N}(\boldsymbol{0}, \boldsymbol{A}^{-1}).$$

$\square$

## B    EXAMPLES

In this section, we discuss the additional details for setting up the numerical examples and show additional examples. In all examples, we use fairly complex covariance matrices, i.e., state-dependent matrices, in order to showcase the effectiveness of our learning. The drift and noise estimations are carried out in both basis method and deep learning method with 3 and 2 being loss functions for estimating drift and covariance, respectively. The observations, serving as the input dataset for testing our method, are generated by the Euler-Maruyama scheme Higham (2001), utilizing the drift functions as we just mentioned. The basis space $\mathcal{H}$ is constructed employing either B-spline or piecewise polynomial methods for maximum degree p-max equals 2. For higher order dimensions where $d \geq 2$, each basis function is derived through a tensor grid product, utilizing one-dimensional basis defined by knots that segment the domain in each dimension.

The parameters will be specified in each subsection of examples. The estimation results are evaluated using several different metrics. We record the noise terms, $\mathrm{d}\mathbf{w}_t$, from the trajectory generation process and compare the trajectories produced by the estimated drift functions, $\hat{\boldsymbol{f}}$, under identical noise conditions. We examine trajectory-wise errors using equation $\rho(\mathbf{x}) = \mathbb{E}\left[\frac{1}{T}\int_{t=0}^{T}\delta_{\mathbf{x}_t}(\mathbf{x})\right]$ with relative trajectory error and plot both $\boldsymbol{f}$ and $\hat{\boldsymbol{f}}$ to calculate the relative $L^2(\rho)$ error using 6, where $\rho$ is derived by equation **??**. When plotting, trajectories with different initial conditions are represented by distinct colors. In trajectory-wise comparisons, black solid lines depict the true trajectories, while blue dashed lines represent those generated by the estimated drift functions. Additionally, the empirical measure $\rho$ is shown in the background of each 1d plot. Furthermore, we assess the distribution-wise discrepancies between observed and estimated results, computing the Wasserstein distance at various time steps with equation 8.

### B.1    EXAMPLE: BENCHMARK MODEL

We consider an SDE model with state dependent noise matrix, as follows

$$\begin{cases} \mathrm{d}\mathbf{x}_t &= C_1\mathbf{x}_t\,\mathrm{d}t + \sqrt{\mathbf{y}_t}\mathbf{x}_t\,\mathrm{d}\mathbf{b}_t^x \\ \mathrm{d}\mathbf{y}_t &= C_2(C_3 - \mathbf{y}_t)\,\mathrm{d}t + C_4\sqrt{\mathbf{y}_t}\,\mathrm{d}\mathbf{b}_t^y, \end{cases}$$

where $(\mathbf{x}_t, \mathbf{y}_t)$ is the pair of state-variables, $(\mathbf{b}_t^x, \mathbf{b}_t^y)$ are standard Brownian motion, the constants $C_1, C_2, C_3, C_4 > 0$ are model parameters. If $2C_2C_3 > C_4^2$, then $\mathbf{y}_t$ remains strictly positive. We use this benchmarking model to test the effectiveness of our learning framework on identifying

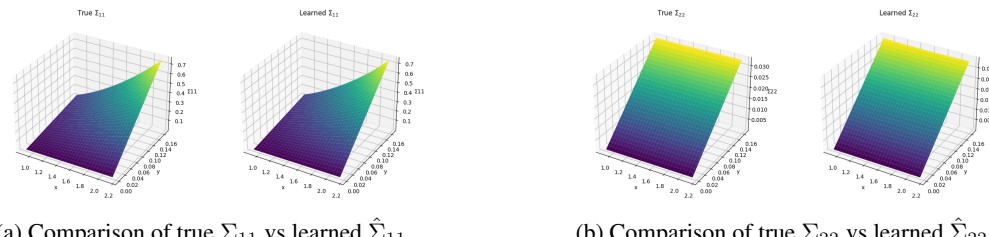

(a) Comparison of true $\Sigma_{11}$ vs learned $\hat{\Sigma}_{11}$.

(b) Comparison of true $\Sigma_{22}$ vs learned $\hat{\Sigma}_{22}$.

Figure 8: Benchmark model: $\Sigma$ vs $\hat{\Sigma}$.

the SDE without any knowledge of the noise and drift terms. We evaluated our learning method on the benchmark model. Trajectories were simulated using the parameters $C_1 = 0.5$, $C_2 = 3.0$, $C_3 = 0.04$, and $C_4 = 0.45$. Both the drift function $f(x,y) = [f_1(x), f_2(y)]$, where $f_1(x) = C_1 x$ and $f_2(y) = C_2(C_3 - y)$, and the diffusion matrix $\sigma(x,y) = \begin{bmatrix} \sqrt{y}x & 0 \\ 0 & C_4\sqrt{y} \end{bmatrix}$ were learned using the neural network method described in the previous section 3.5.

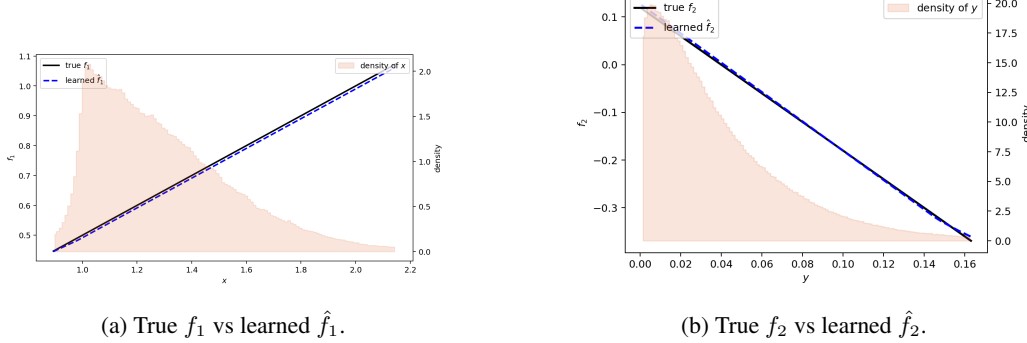

(a) True $f_1$ vs learned $\hat{f}_1$.

(b) True $f_2$ vs learned $\hat{f}_2$.

Figure 9: $\boldsymbol{f}$ vs $\hat{\boldsymbol{f}}$ with empirical distribution of $\mathbf{x}_t$ is shown in the background.

The results are shown in Figure 8, 9 and 10.

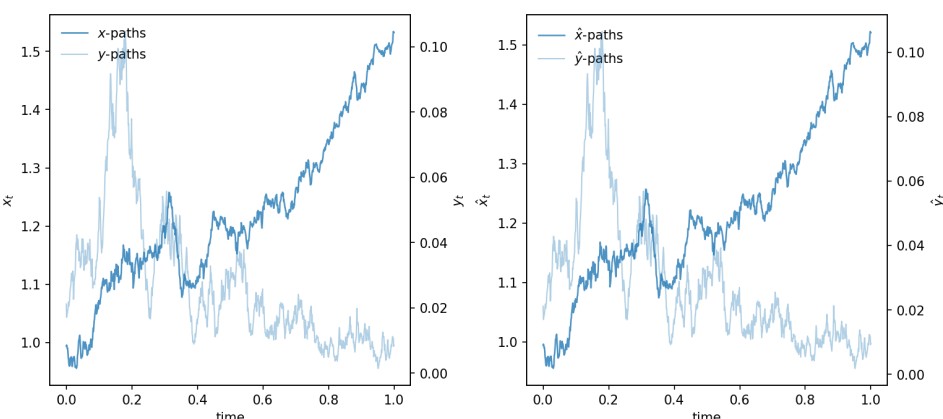

Figure 10: Trajectory comparison with matched noise $db_t$. Left: true simulated paths $(x_t, y_t)$ under the benchmark parameters. Right: re–simulated paths using the learned drift $\hat{f}$ and diffusion $\hat{\sigma}$, driven by the same $(db_t^x, db_t^y)$.

**Conclusion**: By using deep neural networks as the underlying function spaces, one can easily infer those multi-variate drift and noise functions, without specifying the actual form of the functions.

### B.2 EXAMPLE: INTERACTING PARTICLE SYSTEMS (IPS)

If we use the vectorized notations, i.e.

$$\mathbf{x} = \begin{bmatrix} \mathbf{x}_1 \\ \vdots \\ \mathbf{x}_N \end{bmatrix} \quad \text{and} \quad \mathbf{w} = \begin{bmatrix} \mathbf{w}_1 \\ \vdots \\ \mathbf{w}_N \end{bmatrix} \in \mathbb{R}^{D=Nd},$$

and

$$\boldsymbol{f}_\phi(\mathbf{x}) = \begin{bmatrix} \frac{1}{N} \sum_{j=2}^N \phi(\|\mathbf{x}_j - \mathbf{x}_1\|)(\mathbf{x}_j - \mathbf{x}_1) \\ \vdots \\ \frac{1}{N} \sum_{j=1}^{N-1} \phi(\|\mathbf{x}_j - \mathbf{x}_N\|)(\mathbf{x}_j - \mathbf{x}_N) \end{bmatrix}, \quad \sigma = \begin{bmatrix} \sigma^{\mathrm{x}}(\mathbf{x}_1) & \mathbf{0} & \cdots & \mathbf{0} \\ \mathbf{0} & \sigma^{\mathrm{x}}(\mathbf{x}_2) & \cdots & \mathbf{0} \\ \vdots & \vdots & \ddots & \vdots \\ \mathbf{0} & \mathbf{0} & \cdots & \sigma^{\mathrm{x}}(\mathbf{x}_N) \end{bmatrix}.$$

Here each $\mathbf{0}$ is a $d \times d$ matrix, $\boldsymbol{f}: \mathbb{R}^D \to \mathbb{R}^D$ and $\tilde{\sigma}: \mathbb{R}^D \to \mathbb{R}^{D \times D}$. Then the system can be put into one single SDE of the form $\mathrm{d}\mathbf{x}_t = \boldsymbol{f}(\mathbf{x}_t)\,\mathrm{d}t + \tilde{\sigma}(\mathbf{x}_t)\,\mathrm{d}\mathbf{w}_t$. We will consider a weighted $\ell_2$ inner product for these vectors, i.e. for $\mathbf{u}, \mathbf{v} \in \mathbb{R}^d$ with

$$\mathbf{u} = \begin{bmatrix} \mathbf{u}_1 \\ \vdots \\ \mathbf{u}_N \end{bmatrix}, \quad \mathbf{v} = \begin{bmatrix} \mathbf{v}_1 \\ \vdots \\ \mathbf{v}_N \end{bmatrix}, \quad \mathbf{u}_i, \mathbf{v}_i \in \mathbb{R}^d$$

then

$$\langle \mathbf{u}, \mathbf{v} \rangle_N = \frac{1}{N} \sum_{i=1}^N \langle \mathbf{u}_i, \mathbf{v}_i \rangle, \quad \|\mathbf{u}\|_N^2 = \langle \mathbf{u}, \mathbf{u} \rangle_N.$$

With this new norm, we can carry out the learning as usual in $\mathbb{R}^d$ yet with a lower dimensional structure for $\boldsymbol{f}_\phi$ and $\sigma^{\boldsymbol{x}}$. With this setup, the loss of the noise in equation 2 will become

$$\mathcal{E}_\sigma(\tilde{\Sigma}) = \mathbb{E}\Big[\frac{1}{N} \sum_{i=1}^N \big([\mathbf{x}_{i,t}, \mathbf{x}_{i,t}]_0^T - \int_{t=0}^T (\tilde{\sigma}^{\boldsymbol{x}}(\mathbf{x}_{i,t}))^2 \, \mathrm{d}t\big)^2\Big],$$

where we learn $\tilde{\Sigma}^{\boldsymbol{x}} = (\tilde{\sigma}^{\boldsymbol{x}})^2$ as one single SPD matrix using the Cholesky decomposition method described in section 3.5, and then take $\tilde{\sigma}^{\boldsymbol{x}} = \sqrt{\tilde{\Sigma}^{\boldsymbol{x}}}$. Next, the loss of the drift will become

$$\mathcal{E}_{\mathcal{H}}(\varphi) = \frac{1}{2}\mathbb{E}\Big[\int_{t=0}^T \langle \boldsymbol{f}_\varphi(\mathbf{x}_t), \Sigma^\dagger(\mathbf{x}_t)\boldsymbol{f}_\varphi(\mathbf{x}_t)\rangle_N \, \mathrm{d}t - 2\langle \boldsymbol{f}_\varphi(\mathbf{x}_t), \Sigma^\dagger(\mathbf{x}_t)\,\mathrm{d}\mathbf{x}_t\rangle_N\Big].$$

The two terms with the weighted $\ell_2$ inner product can be rewritten as

$$\langle \boldsymbol{f}_\varphi(\mathbf{x}_t), \Sigma^\dagger(\mathbf{x}_t)\boldsymbol{f}_\varphi(\mathbf{x}_t)\rangle_N = \frac{1}{N^3} \sum_{i,j,k=1}^N \varphi(r_{i,j,l}^m)\varphi(r_{i,k,l}^m)\langle \boldsymbol{r}_{i,j,l}^m, (\tilde{\sigma}^{\boldsymbol{x}}(\mathbf{x}_{i,l}^m))^{-2}\boldsymbol{r}_{i,k,l}^m\rangle$$

and

$$\langle \boldsymbol{f}_\varphi(\mathbf{x}_t), \Sigma^\dagger(\mathbf{x}_t)\,\mathrm{d}\mathbf{x}_t\rangle_N = \frac{1}{N^2} \sum_{i,j=1}^N \varphi(r_{i,j,l}^m)\langle \boldsymbol{r}_{i,j,l}^m, (\tilde{\sigma}^{\boldsymbol{x}}(\mathbf{x}_{i,l}^m))^{-2}(\mathbf{x}_{i,l+1}^m - \mathbf{x}_{i,l}^m)\rangle,$$

where $\mathbf{x}_{i,l}^m = \mathbf{x}_i^m(t_l)$, $\boldsymbol{r}_{i,j,l}^m = \boldsymbol{x}_j^m(t_l) - \boldsymbol{x}_i^m(t_l)$, and $r_{i,j,l}^m = \|\boldsymbol{r}_{i,j,l}^m\|$. In estimating $\phi$, we use $[r_{\min}, r_{\max}]$ as the domain for estimation with $r_{ij} = \|\mathbf{x}_j - \mathbf{x}_i\|$ represents the pairwise distance. We use a piecewise local B-spline basis of order up to 3 on domain $[r_{\min}, r_{\max}]$. Let $\mathcal{H} = \mathrm{span}(\{\psi_\eta\}_{\eta=1}^n)$ denote the associated compactly supported basis functions. Then an estimator $\varphi \in \mathcal{H}$ has the form, i.e., $\varphi(r) = \sum_{\eta=1}^n a_\eta \psi_\eta(r)$. For $\mathbf{x} = (\mathbf{x}_1^\top, \ldots, \mathbf{x}_N^\top)^\top$, define interaction features indexed by the particle $i = 1, \ldots, N$, then each $\eta^{th}$ column of $\Phi$ is given as

$$\big(\Phi_\eta(\boldsymbol{x})\big)_i = \frac{1}{N} \sum_{j=1, j\neq i}^N \psi_\eta(r_{i,j}) \, (\sigma^{\boldsymbol{x}}(\boldsymbol{x}_i))^{-1}\boldsymbol{r}_{i,j} \in \mathbb{R}^d,$$

where $\mathrm{v}_{i,j} = \boldsymbol{x}_j - \boldsymbol{x}_i$ and $r_{i,j} = ||\boldsymbol{r}_{i,j}||$. For $M$ trajectories $\{\mathbf{x}_l^m\}_{l,m=1}^{L,M}$, set $\Delta\mathbf{x}_l^m = \mathbf{x}_{t_{l+1}}^m - \mathbf{x}_{t_l}^m$. Then the loss function reduces to

$$\mathcal{E}(a) = \tfrac{1}{2}\, a^\top A\, a - b^\top a,$$

where

$$\begin{cases} A_{ij} & := \frac{1}{M} \sum_{m,l} \langle \Phi_i(\mathbf{x}_l^m),\ \Phi_j \rangle\, \Delta t, \\ b_i & := \frac{1}{M} \sum_{m,l} \langle \Phi_i(\mathbf{x}_l^m),\ \sigma^{-1}(\mathbf{x}_l^m)\Delta\mathbf{x}_l^m \rangle. \end{cases}$$

The estimator is the solution of the normal equations $A\,\hat{a} = b$, hence $\hat{\phi}(r) = \sum_{i=1}^n \hat{a}_i\, \psi_i(r)$.

### B.3 Example: SPDE estimation

For any $N \in \mathbb{N}$, let $H^N = \mathrm{span}\{h_1, \dots, h_N\}$ and $P^N \colon H \to H^N$ the projection operator. Then denote $\mathbf{u}^N = P^N\mathbf{u} = \sum_{k=1}^N \mathbf{u}_k(t)h_k(\mathbf{x})$ as the Fourier approximation of the solution $\mathbf{u}$ by the first $N$ eigenmodes $\mathbf{u}_k(t) = (\mathbf{u}(t), h_k)_H$. The projected solution $\mathbf{u}^N$ of equation 9 satisfies the following finite–dimensional SDE

$$\mathrm{d}\mathbf{u}^N(t, \mathbf{x}) = P^N\big(\theta(\mathbf{x})\,\Delta\mathbf{u}(t, \mathbf{x})\big)\ \mathrm{d}t + \sigma P^N\,\mathrm{d}\mathbf{w}(t, \mathbf{x}). \tag{13}$$

Since eigenmodes are coupled together in term $\theta(x)\Delta\mathbf{u}(t, \mathbf{x})$, $P^N$ does not commute with $\theta(x)$, and to overcome this we consider a Galerkin type projection, i.e.

$$\tilde{\mathbf{u}}^N(t, \mathbf{x}) = \sum_{k=1}^N \tilde{\mathbf{u}}_k(t)h_k(\mathbf{x}) \quad \approx \sum_{k=1}^\infty \mathbf{u}_k(t)h_k(\mathbf{x}) = \mathbf{u}(t, \mathbf{x}),$$

and we have

$$\mathrm{d}\tilde{\mathbf{u}}^N(t, \mathbf{x}) = P^N\big(\theta(\mathbf{x})\,\Delta\tilde{\mathbf{u}}^N(t, \mathbf{x})\big)\ \mathrm{d}t + \sigma\, P^N\,\mathrm{d}\mathbf{w}(t, \mathbf{x}), \tag{14}$$

that we write in a matrix form,

$$\mathrm{d}\tilde{\mathbf{U}}^N(t) = -C_N(\theta)\Lambda_N\,\tilde{\mathbf{U}}^N(t)\ \mathrm{d}t + \sigma Q_N\,\mathrm{d}\mathbf{w}^N(t), \tag{15}$$

where

$$\tilde{\mathbf{U}}^N = \big(\tilde{\mathbf{u}}_1, \dots, \tilde{\mathbf{u}}_N\big)^\mathsf{T}, \tag{16}$$
$$\Lambda_N = \mathrm{diag}(\lambda_1, \dots, \lambda_N), \tag{17}$$
$$Q_N = \mathrm{diag}(q_1, \dots, q_N), \tag{18}$$

and where the matrix $C_N(\theta) \in \mathbb{R}^{N \times N}$ has entries

$$[C_N(\theta)]_{jk} = \langle \theta(x)h_k, h_j \rangle, \qquad 1 \le j, k \le N.$$

Choose any finite dimensional function space $\mathcal{H}$ with basis $\{\psi_i\}_{i=1}^n$, and approximate $\theta(x)$ with respect to this basis,

$$\theta(x) \approx \sum_{i=1}^n a_i\, \psi_i(x), \qquad \boldsymbol{a} = (a_1, \dots, a_n)^\mathsf{T} \in \mathbb{R}^n.$$

For each $i$ define the deterministic matrices

$$[B_N^{(i)}]_{jk} = \langle \psi_i h_k, h_j \rangle, \qquad 1 \le j, k \le N, \tag{19}$$

so that

$$C_N(\theta) = \sum_{i=1}^n \boldsymbol{a}_i\, B_N^{(i)}. \tag{20}$$

Let $\Sigma = \sigma^2 Q_N Q_N^\mathsf{T}$. By our method in section 3,

$$\mathcal{E}(\boldsymbol{a}) = \frac{1}{2}\int_0^T \tilde{\mathbf{U}}^{N\mathsf{T}}\Lambda_N C_N(\boldsymbol{a})\Sigma^{-1}C_N(\boldsymbol{a})\Lambda_N\tilde{\mathbf{U}}^N\ \mathrm{d}t + \int_0^T \tilde{\mathbf{U}}^{N\mathsf{T}}\Lambda_N C_N(\boldsymbol{a})\Sigma^{-1}\ \mathrm{d}\tilde{\mathbf{U}}^N. \tag{21}$$

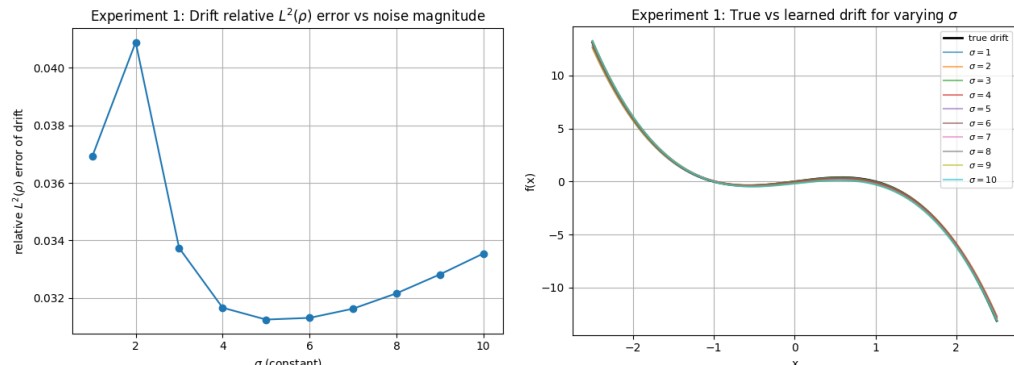

Figure 11: Experiment 1. Left: relative $L^2(\rho)$ error of the drift vs. constant diffusion level $\sigma$. Right: true drift $\boldsymbol{f}_\star$ and learned drifts $\hat{\boldsymbol{f}}_\sigma$ for several values of $\sigma$.

With the expansion defined by (20), we obtain a mass matrix $A$ and RHS vector $\boldsymbol{b}$ having entries given as

$$A_{mk} = \int_0^T \tilde{\mathbf{U}}^{N^\mathsf{T}} \Lambda_N B_N^{(m)} \Sigma^{-1} B_N^{(k)} \Lambda_N \tilde{\mathbf{U}}^N \, \mathrm{d}t, \quad \boldsymbol{b}_m = \int_0^T \tilde{\mathbf{U}}^{N^\mathsf{T}} \Lambda_N B_N^{(m)} \Sigma^{-1} \, \mathrm{d}\tilde{\mathbf{U}}^N,$$

for $1 \le m, k \le n$. $A$ is apparently symmetric positive definite. Next, the loss becomes

$$\mathcal{E}(\boldsymbol{a}) = \frac{1}{2}\boldsymbol{a}^\mathsf{T} A \boldsymbol{a} + \boldsymbol{b}^\mathsf{T}\boldsymbol{a},$$

thus minimizing the loss is equivalent to solving solving the linear system $\nabla\mathcal{E}(\boldsymbol{a}) = 0$, which gives the estimation coefficient as $\hat{\boldsymbol{a}} = -A^{-1}\boldsymbol{b}$.

## C  SUPPORTING EXPERIMENTS

In this appendix we report three $1D$ experiments and one $2D$ experiment designed to directly address the reviewers' concerns on $(i)$ robustness to stochastic noise magnitude, $(ii)$ the effect of observation noise, $(iii)$ the effect in learning time gap (between the actual learning time instances and integration time instances), and $(iv)$ a correlated state dependent noise structure where we show the effects of using a learned noise diffusion matrix as well as a comparion to traditional methods. For the $1D$ cases, we consider the following examples

$$\mathrm{d}\mathbf{x}_t = \boldsymbol{f}_\star(\mathbf{x}_t) \, \mathrm{d}t + \sigma_\star(\mathbf{x}_t) \, \mathrm{d}\mathbf{w}_t, \qquad \boldsymbol{f}_\star(\mathbf{x}) = \mathbf{x} - \mathbf{x}^3,$$

simulate trajectories by Euler–Maruyama with time step $\delta t = 10^{-3}$ on $[0, T]$ with $T = 10$, and use $M = 500$ independent trajectories. The drift is learned in the polynomial space using the discrete version of our noise-aware drift loss, and the error is measured in relative $L^2(\rho)$-norm.

### C.1  EXPERIMENT 1: VARYING THE DIFFUSION MAGNITUDE

The goal of the first experiment is to test how the noise-aware drift estimator behaves as the dynamical noise level varies. We fix

$$\sigma_\star(x) = \sigma \in \{1, 2, \ldots, 10\},$$

simulate $\mathbf{x}_t$ with the true drift $\boldsymbol{f}_\star$ and diffusion $\sigma_\star$. In the drift loss we treat $\sigma^2$ as known and plug in the true value. For each $\sigma$ we compute the learned drift $\hat{\boldsymbol{f}}_\sigma$ and its relative $L^2(\rho)$ error.

### C.2  EXPERIMENT 2: OBSERVATION NOISE

The second experiment investigates the effect of observation nois. We fix a smooth state-dependent diffusion,

$$\sigma_\star(\mathbf{x}) = 0.5 + 0.2\sin(\mathbf{x}),$$

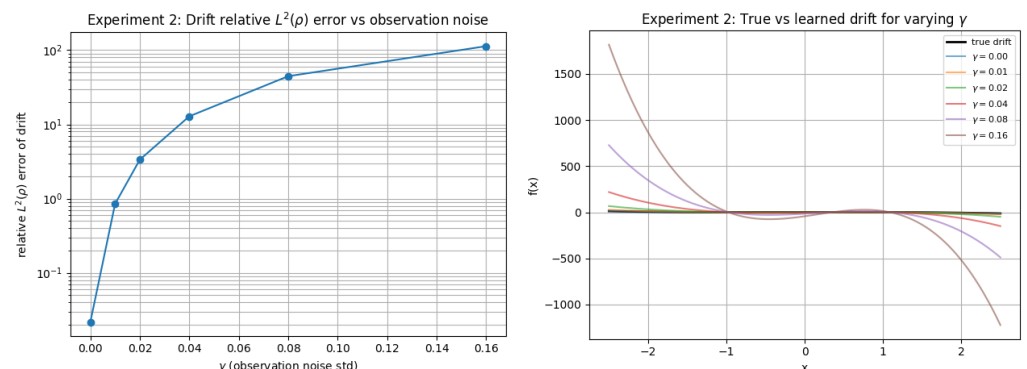

Figure 12: Experiment 2. Left: relative $L^2(\rho)$ drift error vs. observation-noise level $\gamma$. Right: true drift $\boldsymbol{f}_\star$ and learned drifts $\hat{\boldsymbol{f}}_\gamma$ for several values of $\gamma$.

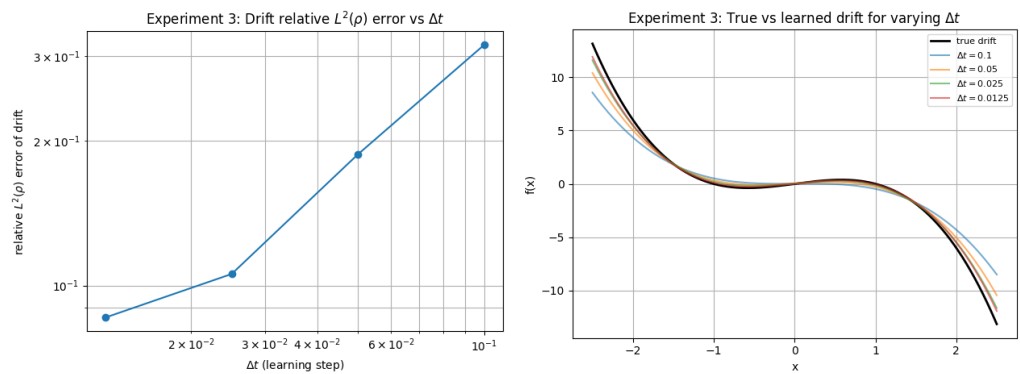

Figure 13: Experiment 3. Left: relative $L^2(\rho)$ drift error vs. learning step $\Delta t$. Right: True drift $\boldsymbol{f}_\star$ and learned drifts $\hat{\boldsymbol{f}}_{\Delta t}$ for several values of the learning step $\Delta t$.

and simulate the true process $\mathbf{x}_t$ as before. On a learning grid with $\Delta t = 10^{-3}$ we form clean states $\mathbf{x}_{t_\ell}$ and then noisy observations

$$\mathbf{y}_{t_\ell} = \mathbf{x}_{t_\ell} + \varepsilon_\ell, \qquad \varepsilon_\ell \sim \mathcal{N}(0, \gamma^2),$$

with independent observation noise. In the loss we assume the diffusion is known and plug in $\sigma_\star^2(\mathbf{y}_{t_\ell})$. We then learn the drift from the noisy increments $\Delta \mathbf{y}_{t_\ell} = \mathbf{y}_{t_{\ell+1}} - \mathbf{y}_{t_\ell}$.

This experiment illustrates that our estimator is designed for stochastic noise for SDE. And dealing with observation noise is filtering problem which is out of the scope of this paper.

## C.3 EXPERIMENT 3: EFFECT OF THE LEARNING TIME STEP $\Delta t$

The third experiment studies the impact of sampling step size. We fix a constant diffusion

$$\sigma_\star(x) \equiv 2,$$

simulate with step size $\delta t = 10^{-3}$, and then subsample the trajectories on learning grids with

$$\Delta t \in \{0.1, \ 0.05, \ 0.025, \ 0.0125\}.$$

Together, these three 1D experiments presents the theoretical properties of our estimator: (i) robustness to changes in the intrinsic noise magnitude; (ii) the expected sensitivity to observation noise, which is not included in our research target; and (iii) different sampling size being consistent with the discrete-time approximation of the continuous-time loss.

## C.4 EXPERIMENT 4: 2D VAN DER POL SDE WITH CORRELATED STATE-DEPENDENT DIFFUSION

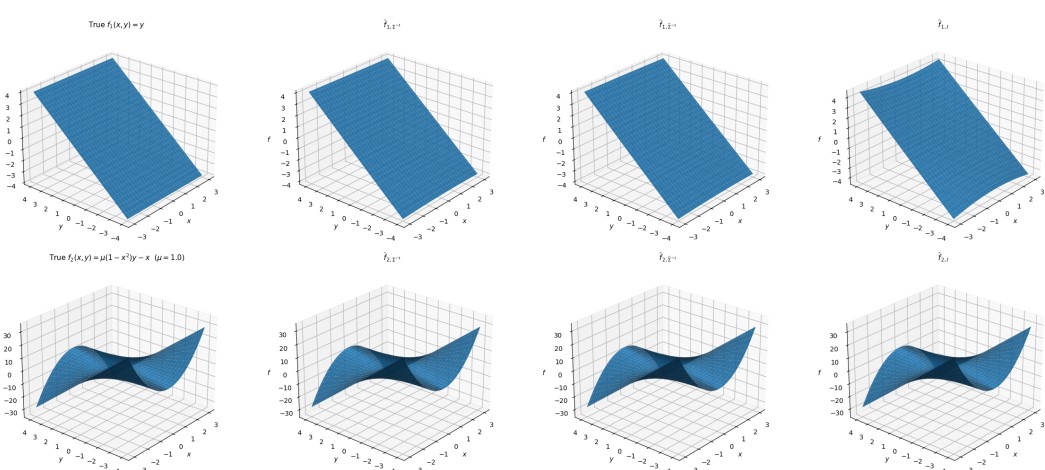

Figure 14: Experiment 4 (2D Van der Pol with correlated, state-dependent diffusion). Top row: true $\boldsymbol{f}_1(\mathbf{x}, \mathbf{y}) = \mathbf{y}$ and learned $\boldsymbol{f}_1$ obtained with $\Sigma_\star^{-1}$, $\widehat{\Sigma}^{-1}$, and the SINDy-like baseline $\Sigma_\star = I$. Bottom row: true $\boldsymbol{f}_2(\mathbf{x}, \mathbf{y}) = \mu(1 - \mathbf{x}^2)\mathbf{y} - \mathbf{x}$ and the corresponding learned $\boldsymbol{f}_2$ for the same three choices.

We also include a 2D example based on the Van der Pol oscillator with a fully non-diagonal, state-dependent diffusion matrix. The drift is

$$\boldsymbol{f}_\star(\mathbf{x}, \mathbf{y}) = \begin{pmatrix} \mathbf{y} \\ \mu(1 - \mathbf{x}^2)\mathbf{y} - \mathbf{x} \end{pmatrix}, \qquad \mu = 1,$$

and we choose a volatility matrix $\sigma_\star(\mathbf{x}, \mathbf{y}) \in \mathbb{R}^{2 \times 2}$ of the form

$$\Sigma_\star(\mathbf{x}, \mathbf{y}) := \sigma_\star(\mathbf{x}, \mathbf{y})\sigma_\star(\mathbf{x}, \mathbf{y})^\top = \begin{pmatrix} v_1(\mathbf{x})^2 & \rho(\mathbf{x}, \mathbf{y})\, v_1(\mathbf{x})\, v_2(\mathbf{y}) \\ \rho(\mathbf{x}, \mathbf{y})\, v_1(\mathbf{x})\, v_2(\mathbf{y}) & v_2(\mathbf{y})^2 \end{pmatrix},$$

with

$$v_1(\mathbf{x}) = 0.1 + 0.03\,\mathbf{x}^2, \qquad v_2(\mathbf{y}) = 0.2 + 0.04\,\mathbf{y}^2, \qquad \rho(\mathbf{x}, \mathbf{y}) = 0.5\,\tanh(0.2\,\mathbf{xy}),$$

so that $\Sigma_\star(\mathbf{x}, \mathbf{y})$ is smooth, positive definite, and non-diagonal. We simulate $M = 500$ trajectories on $[0, 1]$ with time step $\delta t = 10^{-3}$.

We compare our noise aware learning with existing SINDy-like regression methods where $\Sigma_\star = I$ being an unweighted least-squares loss, see section 1.1 for details. This is exactly the structure used in traditional SINDy-type drift estimators, which ignore the correlated, state-dependent covariance.

## D REPRODUCIBILITY STATEMENT

We have taken several steps to ensure the reproducibility of our results.

- The convergence theorem is accompanied by complete proofs in the appendix.
- All algorithms are described in detail with hyperparameters, training procedures, and evaluation metrics fully specified either in the example section or in the additional details of exmaple section in appendix.
- We will provide open-source code, along with scripts to reproduce the experiments, preprocessed datasets (or instructions to obtain them), and random seeds for training, once this paper is accepted.

Together, these measures ensure that independent researchers can reliably reproduce and validate our findings.

# E USE OF THE LLM STATEMENT

We did not employ large language models (LLMs) in the development of this work, including the design of methods, theoretical results, experiments, or analysis. The manuscript was written entirely by the authors, with the exception of occasional use of automated grammar and spelling checkers to improve readability.

