# OpenReview forum: "Noise-Aware System Identification for High-Dimensional Stochastic Dynamics"
_ICLR.cc/2026/Conference — Submitted to ICLR 2026_

### Official Review · Reviewer_LZm2 · 2025-10-19

**Soundness:** 3
**Presentation:** 3
**Contribution:** 1
**Rating:** 2
**Confidence:** 2

**Summary:**

This paper presents a mathematical framework for identifying both the drift and diffusion terms of stochastic differential equations directly from trajectory data. The authors derive a likelihood-based loss for the drift and a quadratic-variation loss for the diffusion, leading to a consistent and asymptotically normal estimator. They extend the approach to high-dimensional settings by parameterizing these functions with neural networks and demonstrate its performance on synthetic SDEs, interacting particle systems, and stochastic PDEs.

**Strengths:**

•	Solid theoretical foundation grounded in stochastic analysis.
	•	Clear derivations linking the loss to likelihood and Radon–Nikodym principles.
	•	Implementation details are transparent and reproducible.
	•	Numerical tests validate correctness and convergence rates.

**Weaknesses:**

•	The work is primarily a statistical estimation study, not a machine-learning or representation-learning contribution.
	•	“Deep learning” (Sec. 3.5) is overstated: networks act only as generic function approximators, not as part of a new ML method.
	•	Experiments are fully synthetic and demonstrate mathematical correctness rather than generalization or learning capability.
	•	Lacks any real or learned high-dimensional data relevant to ICLR (e.g., diffusion generative models, Neural SDEs).
	•	No discussion of computational scaling, sample complexity, or robustness to discrete/noisy observations.
	•	The connection to modern diffusion modeling or score-based generative learning—the key SDE context within ML—is missing.

**Questions:**

1.	How does this framework relate to Neural SDEs or score-based diffusion models used in generative learning?
	2.	How robust is the approach when data are available only at discrete and noisy time points?

---

> ### Author Response · Authors · 2025-12-04
>
> Weakness 1: We disagree. While some of the ideas at high level are rooted in statistical methodologies (loss function based on likelihood type ratio), the gist of the developed methods are based on questions and solutions from AI, machine learning and scientific computing.
>
> Weakness 2: As in the case of AI for Sciences, all deep learning methods are used indeed as function approximators.  Unless we enforce more physics properties to the estimators, then they become physics-informed.
>
> Weakness 3: We use synthetic systems so that the true $(f,\Sigma)$ are known and estimation error can be evaluated, which is standard in SDE/SPDE inference. A full real-data study (hypothesis testing, confidence intervals, measurement noise) is substantial and we view it as follow-up work.
>
> Weakness 4: Our high-dimensional examples are interacting particles and SPDEs, where dimension comes from many degrees of freedom.
>
> Weakness 5: The robustness and noisy observation is explained in response to question 2.
>
> Weakness 6: Score based diffusion and Neural SDE models use SDEs as latent generators for complex data, however we focus on identifying physically meaningful $f$ and $\Sigma$ from trajectories.
>
> Q1: Neural SDEs and score-based diffusion models primarily use SDEs as flexible latent generators for complex data, with objectives such as likelihood maximization or score matching. Our work instead targets operator identification: given state trajectories from a (possibly high-dimensional) physical system, we estimate $f$ and $\Sigma$ via an explicit $\Sigma^{-1}$-weighted negative log-likelihood, and prove convergence in finite-dimensional hypothesis classes.
>
> Q2: While the underlying framework assumes continuous time observations, all experiments and implemented formulas are in a discrete time setup, with a small enough time step $\Delta t$. All approximations are to some extent standard: quadratic variations  are approximated by the limiting sum from the definition of the quadratic variation, while all integrals entering the MLE are either Lebesgue (Riemann), or stochastic integrals approximated by standard sums. Convergence of these quantities as $\Delta \to 0$ are well understood, cf. Kutoyants (2004) for SDEs and Cialenco (2019) for the SPDEs. However, we agree that providing a brief discussion about performance of the proposed estimators in terms of $\Delta$ will further enhance and the manuscript, which we added in Appendix C.

---

### Official Review · Reviewer_XqkQ · 2025-10-29

**Soundness:** 3
**Presentation:** 3
**Contribution:** 2
**Rating:** 4
**Confidence:** 4

**Summary:**

This paper introduces a noise-aware framework for identifying both the deterministic drift and the stochastic diffusion terms in high-dimensional stochastic dynamical systems directly from trajectory data. Unlike methods that treat noise as a nuisance, it jointly learns the full state-dependent and correlated noise structure alongside the drift, using a two-stage approach based on quadratic variation for the diffusion and a likelihood-based loss derived from the Girsanov theorem for the drift. The method is validated on examples like interacting particle systems and stochastic PDEs, demonstrating its ability to handle complex noise and scale to high dimensions using deep learning.

**Strengths:**

* The general idea is easy to follow
* The proposed method is theoretically grounded and is novel

**Weaknesses:**

* The "Related Works" section (1.1) should be moved to a later part of the paper. Currently, it discusses specific methods and loss functions before the core model and notation have been introduced in Section 2. This disrupts the logical flow and may confuse readers. Positioning it as an independent section after the methodology would provide the necessary context for the comparisons made.

* The theoretical derivation assumes continuous-time observation. In practice, data is discrete, and the method relies on fine time discretization (Δt is small, e.g., 0.001 in examples). Its performance with sparse, irregular, or low-frequency data is not explored and would likely degrade significantly, as approximations for dx_t and quadratic variation become poor.

* The two-stage process is elegant but creates a pipeline error. Any inaccuracies in estimating $\Sigma$ will propagate into and bias the subsequent drift estimation $f$, as the drift loss function depends on $\Sigma^{-1}$. The paper does not analyze the sensitivity of the final result to errors in the first stage.

* All experiments use synthetic data with known ground truth. There is no evaluation on empirical datasets.

**Questions:**

* The authors claim superior performance, but the paper lacks comparisons against established baselines. How does the method quantitatively compare against, for instance, a well-tuned Neural SDE or a recent variant of SINDy for SDEs on your own benchmarks?

* You highlight handling "correlated and state-dependent noise," yet key examples use diagonal (IPS) or additive (SPDE) noise. Can you demonstrate the method's performance on a system with a full, non-diagonal, state-dependent diffusion matrix?

* The loss function for the drift is derived from the Girsanov theorem, which is a known concept in stochastic processes. What is the specific algorithmic novelty here? Is it the joint framework, the specific decoupling of the estimation, or the application to high-dimensional learning with NNs?

---

> ### Author Response · Authors · 2025-12-04
>
> Weakness 1: We agree that the current Section~1.1 could have been place better, and as suggested, we moved the discussion on related work to a separate section after the methodology is introduced.
>
> Weakness 2: While the underlying framework assumes continuous time observations, all experiments and implemented formulas are in a discrete time setup, with a small enough time step $\Delta t$. All approximations are to some extent standard: quadratic variations  are approximated by the limiting sum from the definition of the quadratic variation, while all integrals entering the MLE are either Lebesgue (Riemann), or stochastic integrals approximated by standard sums. Convergence of these quantities as $\Delta \to 0$ are well understood, cf. Kutoyants (2004) for SDEs and Cialenco (2019) for the SPDEs. However, we agree that providing a brief discussion about performance of the proposed estimators in terms of $\Delta$ will further enhance and the manuscript, which we added in Appendix C.3.
>
> Weakness 3: This is a good point, and indeed, the two step approximation procedure may lead to some inaccuracy, however, in our model this errors are negligible as tested by extensive numerical experiments. We added a short remark about this in Appendix C.4
>
> Weakness 4: Our primary goal is to introduce and validate a new noise-aware framework in settings with known ground truth. Applying such models to real data and making some meaningful conclusions  is a nontrivial task, that we defer to future work. Such problems are of different nature, and one has to design proper model validation, goodness-of-fit, and hypothesis framework, which for high-dimensional or infinite systems are difficult problems; cf. Cialenco (2014) \it{Hypothesis testing for stochastic PDEs driven by additive noise} for a study on the simplest hypothesis testing problem for a simple SPDE.
>
> Q1: Our ``superior performance'' claim was intended relative to naive regression type estimators that ignore the noise structure. Theoretically, for nontrivial $\Sigma(x)$ our loss is a $\Sigma(x)^{-1}$ weighted regression that coincides with the negative log-likelihood and is asymptotically efficient, whereas SINDy-style approaches typically ignore state dependence and correlations in the noise. We will add empirical comparisons with a SINDy baseline in our examples.
>
> Q2: Thanks for the suggestion. We agree that an example with correlated and colored spatial noise will enhance the manuscript, and in the revised version we added such example in Appendix C.4.
>
> Q3: We agree that the underlying change of measure theorem is classical. However, as we mentioned in the paper, the main novelties are:
> 	(i) an explicit, closed form, noise-aware drift loss that handles general correlated, state-dependent $\Sigma(x)$;
> 	(ii) a two-stage framework that first learns the full $\Sigma(x)$ from quadratic variation and then uses it in the likelihood-based drift estimation, rather than treating noise as an unmodeled term;
> 	(iii) high dimensional systems (interacting particle systems and SPDEs), where we exploit the structure of the system to obtain efficient linear systems and modeled learning of SPDE into our SDE learning framework; (iv) considering state and space depended drift in contrast to the existing literature where the Girsanov transformation is performed for learning the drift  assuming $\theta$ being constant.

---

### Official Review · Reviewer_Qb5c · 2025-10-29

**Soundness:** 2
**Presentation:** 2
**Contribution:** 1
**Rating:** 2
**Confidence:** 4

**Summary:**

This paper proposes a noise-aware framework for identifying both the drift and diffusion terms in high-dimensional stochastic dynamical systems from trajectory data. The method is derived from the Girsanov theorem and the Radon–Nikodym derivative, leading to a likelihood-based loss that allows simultaneous estimation of the deterministic and stochastic components without assuming the specific form of the noise model. The authors validate their approach on several examples, including interacting particle systems and stochastic PDEs, and provide theoretical convergence guarantees.

**Strengths:**

1. The paper provides a solid derivation of the drift loss based on stochastic process theory, which is different from the existing methods.

2. Demonstrations on both finite-dimensional and PDE-type stochastic systems show good performance of the proposed method.

**Weaknesses:**

1. The paper does not compare with established SDE inference methods such as [R1],[R2],[R3] . Without such benchmarks, it is difficult to assess how much improvement the proposed method offers beyond the existing works.

2. Although the introduction mentions physics, biology, and finance, the experiments are purely toy models. It would be useful to explore whether the method could be applied to financial time series or stock dynamics, where stochastic modeling is central, or to other real data domains such as EEG brain signals.

3. Since the method claims to be noise-aware, it would be important to analyze its behavior under different noise magnitudes or correlated noise.

__References__
[R1] Course, K., & Nair, P. B. (2023). State estimation of a physical system with unknown governing equations. Nature, 622(7982), 261-267.

[R2] Oh, Y., Lim, D. Y., & Kim, S. (2024). Stable neural stochastic differential equations in analyzing irregular time series data. arXiv preprint arXiv:2402.14989.

[R3] Li, X., Wong, T. K. L., Chen, R. T., & Duvenaud, D. (2020, June). Scalable gradients for stochastic differential equations. In International Conference on Artificial Intelligence and Statistics (pp. 3870-3882). PMLR.

**Questions:**

Q1: How does the proposed method compare quantitatively to recent SDE inference frameworks such as the references [R1],[R2],[R3] mentioned in weaknesses.?

Q2: Can the authors demonstrate or discuss whether the learned model generalizes to real-world stochastic processes, for example, financial or biophysical data?

---

> ### Author Response · Authors · 2025-12-04
>
> Q1: [R1] addresses state estimation (filtering/smoothing) with partially unknown dynamics, using a Markov Gaussian process and variational inference; the goal is to reconstruct latent states, not to consistently identify $(f,\Sigma)$ on the full state space. [R2] and [R3] develop neural SDE frameworks and scalable gradient computation for irregular time series and downstream tasks, treating SDEs mainly as flexible latent models. By contrast, our focus is operator identification: deriving an explicit, $\Sigma^{-1}$-weighted, likelihood based loss for $f$ and $\Sigma$, proving convergence in finite-dimensional hypothesis classes, and specializing to structured high-dimensional regimes (interacting agents, SPDEs). Because the problem settings differ (partial observation/irregular sampling vs.\ full-state trajectories with known ground truth), a direct benchmark is not straightforward. We added a numerical comparison with existing SINDy-like method in Appendix C.4.
>
> Q2: Theoretically, our estimator can be applied whenever the underlying dynamics are reasonably modeled by an SDE (e.g., certain financial or biophysical systems): given trajectories of the observed state, we can learn $f$ and $\Sigma$ exactly as in the synthetic examples. The main challenges are in evaluation: there is rarely a known ``true'' SDE. A careful analysis would require hypothesis testing, confidence intervals, and treatment of measurement noise, as is standard in the SDE/SPDE inference literature see Hypothesis testing for stochastic PDEs driven by additive noise by Cialenco (2014).

---

### Official Review · Reviewer_4XYQ · 2025-10-30

**Soundness:** 2
**Presentation:** 3
**Contribution:** 2
**Rating:** 2
**Confidence:** 3

**Summary:**

The paper proposes a new framework for the estimation of SDEs via a novel loss function, which is derived from a suitable negative log-likelihood. The authors derive a convergence result in the case where the estimator belongs to a hypothesis class with finite dimension. They evaluate their approach on two synthetic examples: a system of interacting particles and the stochastic heat equation.

**Strengths:**

- Clearly written and motivated paper
- Theoretical guarantees: convergence results

**Weaknesses:**

- Lack of scalability
- Validation only on synthetic data
- No numerical comparison against the state-of-the-art in learning SDEs
- Numerical evaluation limited to relatively small dimension not matching the claim of scalability
- No evaluation on real-world data

**Questions:**

1/ Could you elaborate on the claim of existence and uniqueness of estimator under your formulation (line 98) ?


2/ Can the theoretical results be maintained if one assumes trajectories that are discrete in time as is the case in practice ?


3/ Since your method requires computing matrix square-root which is cubic in complexity, can you justify your claim of scalability to high-dimensions ?

---

> ### Author Response · Authors · 2025-12-04
>
> Weakness 1: This is an inaccurate  statement. This point has been shown explicitly in in the interactive particle system example, and SPDEs learning example.
>
> Weakness 2: The  primary goal of this paper is to introduce and validate a new noise-aware framework in settings with known ground truth. Applying such models to real data and making some meaningful conclusions  is a nontrivial task, and deferred to future work. Such problems are of different nature, and one has to design proper model validation, goodness-of-fit, and hypothesis testing framework, which for high-dimensional or infinite systems are difficult problems; cf. Cialenco (2014) \it{Hypothesis testing for stochastic PDEs driven by additive noise} for a literature review  and a study on the simplest hypothesis testing problem for a simple SPDE.
>
> Weakness 3: We already showed the theoretical comparison in related methods. For numerical experiment, we compared our method with SINDy-like regression in Appendix C.4
>
> Weakness 4: This is a repeated and an inaccurate statement. See reply to point 1.
>
> Weakness 5: This is another repeated statement. See reply to point 2.
>
> Q1: This is a valid point. In the nutshell, the minimization problem is linear-quadratic, due to the fact that drift is projected to a finite dimensional linear space. This optimization problem has a unique solution, thanks to standard results in optimization theory. We added a remark about this.
>
> Q2: While the underlying framework assumes continuous time observations, all experiments and implemented formulas are in a discrete time setup, with a small enough time step $\Delta t$. All approximations are to some extent standard: quadratic variations  are approximated by the limiting sum from the definition of the quadratic variation, while all integrals entering the MLE are either Lebesgue (Riemann), or stochastic integrals approximated by standard sums. Convergence of these quantities as $\Delta \to 0$ are well understood, cf. Kutoyants (2004) for SDEs and Cialenco (2019) for the SPDEs. However, we agree that providing a brief discussion about performance of the proposed estimators in terms of $\Delta$ will further enhance and the manuscript, which we added in Appendix C.3
>
> Q3: In our framework we learn and use the covariance $\Sigma(x)$ directly; a particular diffusion coefficient $\sigma(x)$ with $\Sigma(x)=\sigma(x)\sigma(x)^\top$ is introduced only for notational convenience when writing the SDE or the associated Fokker--Planck equation, which in fact depends only on $\Sigma$. Thus, our training and simulation pipelines never require computing matrix square-roots of empirical covariances at each step. If we display a specific $\sigma$, this is done offline for illustration only and does not affect the computational complexity in the regimes where we claim scalability.

---

### Meta-Review · Area_Chair_x6Uc · 2025-12-15

**Summary:**

This paper introduces a noise-aware framework for identifying high-dimensional stochastic dynamical systems directly from trajectory data. The proposed method estimates the diffusion term using quadratic variation and the drift term using a loss function derived from the Girsanov theorem and Radon-Nikodym derivative. The authors provide theoretical analysis regarding the consistency and asymptotic behaviors, and numerical results using interacting particle systems and stochastic PDEs.

Strengths of this paper include the sound theoretical foundation and mathematical rigor of the derivation. The main concerns are the limitations in the paper's contribution and evaluation. In particular, a primary concern shared across the board was the lack of significant datasets to justify the practical utility of the method. Comparisons against established baselines were provided during the rebuttal but they were not as comprehensive as needed. As such, the recommendation is for rejection.

**Reviewer Concerns:**

The authors addressed some of reviewers' questions on existence & uniqueness of the estimators, complexity of matrix square root computation, performance under correlated noise and some writing and novelty claims. New numerical results were added to address concerns on the lack of baselines, result soundness in discrete time settings and error propagation during the two-stage process. These results help but the experimental settings were relatively limited.

One major common concern raised by all reviewers is the absence of evaluation on more sophisticated, potentially real-world data. The authors deferred this to future work, citing the need for a rigorous hypothesis testing framework. This is ok to defend the method but leaves the practical robustness of the method unverified and limits its empirical impact.

**Reviewer Scores:**

Reviewers may slightly raise their scores acknowledging the clarifications on the theory and the extra numerical result added. However, the main factor preventing a shift into acceptance would be the lack of evidence that the method would be useful in more realistic settings.

---

### Decision · Program_Chairs · 2026-01-26

Reject